# Intelligent Monitoring and Trend Analysis of Surface Soil Organic Carbon in the Black Soil Region Using Multi-Satellite and Field Sampling: A Case Study from Northeast China

**DOI:** 10.3390/s25175442

**Published:** 2025-09-02

**Authors:** Chaoqun Chen, Huimin Dai, Kai Liu, Yulei Tang

**Affiliations:** 1Shenyang Center of China Geological Survey, Shenyang 110034, China; 2Northeast Geological S&T Innovation Center of China Geological Survey, Shenyang 110034, China; 3Key Laboratory of Black Soil Evolution and Ecological Effect, Ministry of Natural Resources/Liaoning Province, Shenyang 110034, China; 4Center for Geophysical Survey, China Geology Survey, Langfang 065000, China

**Keywords:** soil organic carbon, multi-satellite, XGBoost, Tongken River Basin

## Abstract

**Highlights:**

**What are the main findings?**
Multi-source satellite data (Landsat-9 and GF-1) were synergistically used through an XGBoost-RFECV framework (R2 = 0.9130 and RMSE = 0.3834%).Full-coverage SOC mapping complemented site-specific monitoring, revealing parent material (correlation coefficient = 0.38) as the dominant control.

**What is the implication of the main findings?**
The integrated multi-source remote sensing method offers a practical reference for high-precision, detailed spatiotemporal SOC monitoring in agricultural watersheds.Identifying parent material as the key control factor provides a case reference for developing soil carbon sequestration strategies and sustainable land management in similar regions.

**Abstract:**

The black soil region of northeast China is a critical global grain production base. The dynamic variations in soil organic carbon (SOC) are directly linked to the regional food security. To accurately monitor SOC content and evaluate the potential of integrating Landsat-9 and GF-1 satellite data for SOC inversion, we developed a machine learning framework that combines data from both satellite sources to model SOC. Using the typical black soil region of northeast China in the Tongken River Basin as the study area, we compared the MLR, PLSR, RF, and XGBoost algorithms. And XGBoost demonstrated the highest performance (R^2^ = 0.9130; RMSE = 0.3834%). Based on the optimal model, SOC in the study area was projected from 2020 to 2024. The multi-year average SOC exhibited an initial increase followed by a subsequent decline, with an overall increase of 22.78%. Spearman correlation analysis identified parent material as the dominant factor controlling SOC variation at the watershed scale (correlation coefficient = 0.38) while also modulating the influence of land use types on SOC dynamics. The “space–ground” multi-source collaborative inversion framework developed in this study offers a high-precision technical approach for the monitoring of SOC in black soil regions.

## 1. Introduction

Northeast China is one of the world’s four major black soil regions and a key global grain-producing area. To strengthen the protection of black soil, the Chinese government has enacted the ‘Law of the People’s Republic of China on the Protection of Black Soil.’ According to relevant legal provisions, black soil is defined as soil with a black or dark black humus surface layer, characterized by good structure and high fertility. In the World Reference Base classification, the majority of black soils correspond to chernozems, kastanozems, and phaeozems [1,2]. However, research conducted by the Food and Agriculture Organization of the United Nations indicates that global black soil regions are experiencing a loss of organic carbon at an annual rate of 0.3 to 1.0 percent, a trend especially evident in areas characterized by intensive agricultural practices [3]. Therefore, the establishment of an efficient and accurate soil organic carbon (SOC) monitoring system holds significant scientific value in promoting the sustainable management of black soil resources and ensuring global food security.

Conventional SOC monitoring methods primarily depend on grid-based sampling combined with laboratory analysis. While these approaches can achieve a measurement accuracy of ±0.5%, they often fall short in terms of temporal and spatial resolution, making it challenging to fulfill the demands of dynamic monitoring at the regional scale. With the rapid advancement of remote sensing technology, spectral inversion techniques have emerged as an effective approach for enabling large-scale SOC monitoring across regional scales. As early as 1965, international researchers began to investigate the sensitive spectral bands associated with SOC content and to establish correlations between SOC and spectral characteristics. Research has demonstrated that the visible light to short-wave infrared (SWIR) band exhibits a significant correlation with SOC content, indicating its potential as an effective indicator for SOC assessment [4]. Saha et al. carried out a study utilizing Hyperion hyperspectral satellite data collected from regions in western India. The results of the correlation analysis indicate that the spectral reflectance of Hyperion band 57, corresponding to a wavelength of 1033.88 nm, exhibits the highest correlation with SOC content, with a correlation coefficient of −0.86 [5]. Castaldi et al. conducted a systematic evaluation of the sensitivity and predictive capacity of multispectral satellite sensors in relation to SOC. Their findings revealed that the SWIR and red bands demonstrated the strongest negative correlations with SOC (correlation coefficient < −0.6) [6]. Gholizadeh et al. systematically elaborated on the spectral response mechanism of SOC in their review, proposing that distinct absorption bands exist for the C=O bond in the range of 2100–2300 nm and for the C=H bond in the range of 1650–1750 nm, which represent optimal wavelength ranges for SOC inversion [7]. Therefore, integrating remote sensing spectral data with ground-based auxiliary information can effectively enhance the prediction accuracy of surface SOC, significantly enhance model accuracy, and thereby improve the spatial inversion of regional SOC distribution [8].

In focusing on the research of SOC content inversion, it is essential to closely examine key SOC inversion technology. With the rapid advancement of big data technologies, machine learning has been increasingly applied in the identification of spectral sensitive bands associated with SOC and in quantitative inversion processes. Wang et al. proposed that the Two-Point Machine Learning (TPML) method can effectively address the challenges associated with inverting SOC content from sparse ground sample points, thereby significantly enhancing inversion accuracy [9]. Ma et al. proposed a deep learning model to estimate SOC content from multispectral remote sensing data, which demonstrated significantly higher prediction accuracy compared to traditional methods [10]. Saygn et al. utilized a range of machine learning models to predict various soil properties in Vezirköprü district of Samsun province [11]. Zhang et al. asserted that, irrespectively of the combination of climatic and topographic data employed, extreme gradient boosting (XGBoost) model demonstrated the highest accuracy in estimating SOC content within the western coastal wetlands of Bohai Bay [12].

With the continuous deployment of satellites equipped with various sensors and differing spatial resolutions, the application of multi-source data fusion techniques to soil moisture inversion has emerged as an effective methodology. Existing research indicates that multi-sensor data fusion offers significant advantages over single-source data with respect to data accuracy and practical applicability [13]. Yan et al. demonstrated that utilizing GF-5 hyperspectral imagery combined with Sentinel-1 backscattering coefficients effectively reduces noise interference and significantly improves the inversion accuracy of SOC content, thereby providing technical support for SOC prediction in the central Yunnan Plateau region [14]. Wei achieved high-resolution spatial prediction of SOC in small-scale hilly agricultural regions through the integration of time series multispectral and radar data sourced from various domestic satellite platforms [15].

The Landsat series of satellites, characterized by their multi-spectral bands, have played a crucial role in the spatial mapping and long-term, large-scale monitoring of SOC [16,17]. The most recent satellite in the Landsat series is Landsat-9, which was successfully launched in 2021. Compared to Landsat-8, Landsat-9 exhibits higher contrast and entropy values, suggesting an enhanced capability to detect more subtle surface changes [18]. However, studies investigating the potential and feasibility of utilizing Landsat-9 for the assessment of SOC remain limited [19]. Most existing studies have primarily focused on Landsat-8 or Sentinel-2 data, and a systematic evaluation of how to optimize SOC estimation by leveraging the distinct spectral response characteristics of Landsat-9 remains lacking. The spatial distribution of SOC exhibits significant heterogeneity due to variations in agricultural practices and soil erosion in the northeast black soil region; this heterogeneity imposes limitations on the accuracy of Landsat data when monitoring fragmented farmland types, such as small plots and terraced fields. And at the small-watershed scale, the mixed pixel problem can significantly compromise the accuracy of SOC inversion results. Therefore, the integration of high-resolution satellite spectral data is essential to improve the precision and accuracy of SOC mapping in black soil regions. The GF-1 satellite was launched in 2013 [20]. It has a spatial resolution of 16 m and a revisit cycle of four days. These characteristics make the satellite particularly suitable for precise monitoring of farmland soil [21]. Therefore, the integration of Landsat-9 and GF-1 data for SOC inversion can effectively utilize the complementary strengths of both satellite systems. The high temporal resolution of GF-1 enables the capture of short-term surface changes, thereby compensating for the time series data gaps in Landsat-9 that are caused by cloud cover and precipitation. Meanwhile, the multispectral bands of Landsat-9 offer essential soil spectral information that is not available in GF-1. However, current research on the feasibility of synergistic inversion of SOC using multi-source data from Landsat-9 and GF-1 remains limited, and the potential of this novel data combination paradigm for improving SOC monitoring accuracy requires systematic validation.

Therefore, to investigate the feasibility and potential of integrating Landsat and GF satellite data for SOC retrieval, this study aims to address the following key research questions: Can the integration of Landsat-9 and GF-1 satellite data significantly enhance the accuracy of remote sensing inversion for SOC content in black soil regions? Which machine learning model performs the best? The objectives were threefold: (1) Data fusion of Landsat-9 and GF-1 imagery was performed, with sensitive spectral bands selected by RF-RFECV. Multiple machine learning models (MLR, PLSR, RF, XGBoost) were compared for SOC estimation in black soils. (2) Spatio-temporal mapping was implemented using the optimal model, generating annual SOC distribution maps (16 m × 16 m) for the Tongken River Basin (2020–2024). (3) Dynamic change analysis employed Slope-based trend detection. Based on predicted SOC, we assessed five-year trends and erosion-induced carbon loss. This study provides a case reference for technical methods of multi-source remote sensing data fusion in soil health management and agricultural sustainable development in black soil regions.

## 2. Materials and Methods

### 2.1. Study Area

The study area is situated within the Tongken River Basin, located in the central region of Heilongjiang Province, China (46.51–48.00° N, 125.92–127.72° E). The Tongken River serves as a first-order tributary on the left bank of the Songhua River, originating from the southern slopes of the Xiaoxing’anling Mountains. It flows through several administrative regions, including Suihua City, Hailun City, and Wangkui County, before ultimately discharging into the Hulan River in Beilin District, Suihua City. The total drainage area of the basin is 10,476.8 km^2^ (Figure 1). The northeastern part of the study area consists of low mountains and hilly terrain, which gradually gives way to a plain region in the western direction [22]. The study area is characterized by diverse soil types, predominantly black soil and meadow soil. The soil profile is deep and exhibits a high organic carbon content. The thickness of the soil humus layer generally ranges from 20 to 50 cm, and in certain regions, it may extend to 80 to 100 cm [23]. This region serves as a significant commercial grain production base in Heilongjiang Province, primarily cultivating crops such as corn, soybeans, and rice. The drainage system within the basin exhibits a dendritic pattern, with major tributaries such as the Zayin River and the Hailun River [24]. The Tongken River exhibits a high sediment load, and severe surface erosion contributes to widespread soil erosion [25].

### 2.2. Data Sources and Processing

#### 2.2.1. Satellite Retrievals

Landsat-9 OLI/TIRS and GF-1 WFV images were used as the required remote sensing data in this study. The Landsat-9 satellite remote sensing images were sourced from the Landsat-9 Surface Reflectance Tier 2 dataset in United States Geological Survey (https://earthexplorer.usgs.gov/ accessed on 12 March 2024). All surface reflectance data in this dataset underwent atmospheric correction. GF-1 satellite remote sensing images were sourced from the Geospatial Data Cloud (https://www.gscloud.cn/home accessed on 20 February 2025). Annual Landsat-9 and GF-1 images of the study area from 2020 to 2024 were selected, with data acquisition primarily occurring in March and May. Only images with cloud cover below 2% were selected for further processing. The GF-1 images underwent a series of preprocessing steps, including band combination, radiometric calibration, atmospheric correction, and mosaicking. The Landsat-9 images underwent band combination and mosaicking. To achieve the fusion analysis of multi-source remote sensing imagery, spatial resampling was performed on the Landsat-9 images. The spatial resolution of all datasets was uniformly adjusted to 16 m using the Bicubic Convolution algorithm. On high-resolution GF-1 images, 20 to 30 permanent ground features that exhibit stable temporal and spatial characteristics, such as road intersections and corners of dam structures, were selected as ground control points. A first-order affine transformation was employed to perform cross-sensor geometric registration between Landsat-9 and GF-1. The registration accuracy control standard specifies that the root mean square error (RMSE) should be less than 0.5 GF-1 multispectral pixels, equivalent to 8 m or less. Finally, the first seven bands of Landsat-9 and the four spectral bands of GF-1 were integrated. The temporal information of the selected images is presented in Table 1.

To strengthen the correlation between spectral features and SOC content, this study applied multi-dimensional spectral transformation techniques to Landsat and GF-1 satellite data. The specific methods included the following: reciprocal transformation (1/R), logarithmic transformation (logR), inverse logarithmic transformation (log(1/R)), first-order differentiation (R′), second-order differentiation (R″) [26], logarithmic reciprocal first-order differentiation (d(log(1/R))/dλ), and continuum removal transformation (CR) [27]. All remote sensing image processing tasks and spectral mathematical transformations were conducted using ENVI 5.6 (Exelis Visual Information Solutions, Broomfield, CO, USA).

#### 2.2.2. Soil Sample Collection

In this study, a multi-source data fusion model for SOC in black soil was developed based on the measured SOC data in 2023. Considering the representativeness and spatial heterogeneity of soil in the Tongken River Basin, this study employed a stratified random sampling method based on spatial stratification of land use and elevation to establish sample points in 2023. Through a hierarchical design approach, the entire basin was subdivided into 66 distinct regions. Except for water, three to four sampling points were randomly established in each of these regions. Soil samples were collected from the surface layer (0–20 cm). Following natural air-drying, the samples were sieved using a 20-mesh nylon sieve. The SOC content was determined using the potassium dichromate oxidation method with external heating [28]. Following the exclusion of outliers, a total of 204 soil samples were collected in 2023. The sample points were divided into a training set and a testing set at a ratio of 7:3. To ensure adequate spatial coverage of the validation data, the testing set was preferentially distributed across larger areas. To assess the model’s capability in capturing recent trends and its short-term extrapolation performance, this study employed 60 soil samples from 2022 and 56 soil samples from 2024 to validate the optimal model’s simulated SOC distributions for the corresponding years. The distribution of sampling points is shown in Figure 2 and the statistical information is shown in Table 2.

#### 2.2.3. Geographic Covariates

The formation and accumulation of SOC are influenced by a combination of ecological, geological, environmental, and anthropogenic factors [29,30]. To investigate the primary factors influencing variations in SOC at the basin scale, this study selected 11 indicators across five categories (topography and geology, meteorology, soil characteristics, ecological environment, and human activities) for correlation analysis. The indicators and corresponding data content are presented in Table 3. All datasets used in the study were projected in CGCS2000_GK_CM_129E and resampled to a spatial resolution of 16 m. These operations were conducted using QGIS 3.1.

### 2.3. RF-RFECV Spectral Band Extraction

Since spectral mathematical transformation not only enhances the correlation between reflectance and SOC content but also increases accompanying noise, this paper employed the RF-RFECV machine learning method for sensitive band selection [32,33].

Recursive Feature Elimination Cross Validation (RFECV) is a machine learning-based feature selection technique. It can be characterized as an iterative procedure performed on a base classification model, in which cross-validation is conducted across various feature combinations. The validation errors of all feature subsets are computed using the base model, and the subset yielding the lowest error rate is selected as the optimal feature subset [34]. Owing to the structural properties of decision trees, the Random Forest (RF) model is capable of performing dimensionality reduction when processing high-dimensional feature inputs. Therefore, RF is selected as the supervised learning estimator for RFECV [35]. RF-RFECV machine learning method is capable of effectively processing high-dimensional spectral data with strong inter-variable correlations. The tree structure splitting mechanism in RF does not depend on the assumption of feature independence and inherently exhibits robustness against multicollinearity. By recursively eliminating the least important features, the method ensures that only the most discriminative features are retained within highly correlated feature groups. And ten-fold cross-validation is employed to ensure that the selected feature subset exhibits optimal predictive performance on unseen data. This method was implemented through the development of a Python program using JetBrains PyCharm 2019.1.1 (Community Edition) (JetBrains, Prague, Czech Republic).

### 2.4. Model Development for SOC Content

To establish an optimal monitoring model for black SOC that is well-suited for multi-source data fusion, this study systematically compared a range of statistical and machine learning models in order to identify the most appropriate inversion method. The statistical models considered included multiple linear regression (MLR) and Partial Least-Squares Regression (PLSR), while the machine learning models evaluated encompassed RF and XGBoost. All methods were implemented using JetBrains PyCharm 2019.1.1 (Community Edition) (JetBrains, Prague, Czech Republic).

#### 2.4.1. MLR

Multiple linear regression is a statistical modeling technique employed to examine the linear relationship between a single dependent variable and two or more independent variables [36]. As an extension of simple linear regression, it is widely utilized for both predictive and explanatory purposes in scenarios involving multiple influencing factors. The mathematical formulation is presented as follows:(1)$$Y=\beta_{0}+\beta_{1}X_{1}+\beta_{2}X_{2}+\ldots +\beta_{n}X_{n}$$
where Y is the SOC of black soil, $X_{1},X_{2},\ldots ,X_{n}$ are the independent variables of the model, $\beta_{0}$ is the intercept, and $\beta_{1},\beta_{2},\ldots ,\beta_{n}$ are the regression coefficients, indicating the degree of influence of each independent variable on the dependent variable.

#### 2.4.2. PLSR

PLSR offers more reliable modeling outcomes in situations where independent variables are highly correlated, and it is particularly well-suited for constructing regression models when the number of samples is smaller than the number of independent variables [37]. The calculation approach is as follows: if there are n soil samples, with SOC denoted as Y and p bands of spectral data forming p independent variables ($X=\left\{x_{1},x_{2},\cdots ,x_{p}\;\right\}$), the first principal component $t_{1}$ is derived in such a way that it captures the maximum amount of variation from X. Additionally, $t_{1}$ exhibits the highest correlation with Y, indicating that it has the strongest explanatory power for the variation in Y. After the first principal component $t_{1}$ is extracted, both Y and X are regressed on $t_{1}$. The algorithm terminates when the desired level of equation accuracy is achieved. Otherwise, the second principal component, $t_{2}$, is extracted by utilizing the residual information of X and Y that remains unexplained after $t_{1}$. This iterative procedure is repeated until the desired level of accuracy is achieved [38].

#### 2.4.3. RF

The RF algorithm exhibits favorable interpretability in addressing nonlinear problems, demonstrates robust noise resistance, requires relatively low data completeness, and achieves high predictive accuracy. Consequently, it is extensively utilized in the prediction of soil properties [9,39]. The model was proposed by Leo Breiman in 2001 and is a machine learning algorithm that integrates multiple classification and regression trees [40]. It does not require dimensionality reduction when processing multi-dimensional datasets, and it can preserve the original information of datasets to the greatest extent. The main workflow is as follows: n samples are randomly selected with replacement from the original set of n training samples to form *m* bootstrap subsets. Each subset is then used independently to train a decision tree. The final output of the random forest regression model is obtained by averaging the prediction results from all *m* decision trees.

In this study, hyperparameter optimization of the RF model was conducted using the randomized search method, with performance evaluation of different hyperparameter combinations carried out through 10-fold cross-validation. Following the optimization through the random search, the final set of hyperparameters was determined as follows: the number of trees was 400, the maximum tree depth was 50, the minimum number of samples required for a split was 2, the minimum number of samples required at a leaf node was 1, and the proportion of features considered for splitting was 0.7.

#### 2.4.4. XGBoost

XGBoost is an ensemble tree-based learning algorithm that integrates both bagging and boosting techniques. It utilizes the second-order Taylor expansion of the loss function to approximate the negative gradient, which is subsequently employed as the residual of the preceding model iteration for further learning [41]. The calculation formula is as follows [42]:(2)$$Y_{i}=\sum_{k=1}^{K}f_{k}(x_{i}),f_{k}(x_{i})\in F$$
where $Y_{i}$ is the predicted value of the model, *k* is the number of integrated regression trees, $x_{i}$ is the data sample, F is the function space composed of regression trees, and $f_{k}$ is a specific regression tree within F.

In this study, the hyperparameters of the XGBoost model were optimized using the randomized search method in conjunction with 10-fold cross-validation. The final hyperparameters, determined through random search optimization, were as follows: the number of trees was set to 200, the maximum tree depth was 5, the learning rate was 0.05, the sampling ratio for both samples and features was 0.9, the minimum loss reduction required for node splitting was 0.1, and all regularization coefficients were set to 0.5.

### 2.5. Predictive Performance Evaluation

To verify the accuracy and stability of the inversion model, the coefficient of determination (R^2^), root mean squared error (RMSE), mean absolute error (MAE), and bias were employed as evaluation metrics, with the corresponding formulas presented as follows [43]:(3)$$R^{2}=1-\sum_{i=1}^{n}\frac{{\left(\hat{y_{i}}-\bar{y}\right)}^{2}}{\sum_{i=1}^{n}{\left(y_{i}-\bar{y}\right)}^{2}}$$(4)$$RMSE=\sqrt{\frac{\sum_{i=1}^{n}{\left(\hat{y_{i}}-y_{i}\right)}^{2}}{n}}$$(5)$$MAE=\frac{1}{n}\sum_{i=1}^{n}\left|\hat{y_{i}}-y_{i}\right|$$(6)$$Bias=\frac{1}{n}\sum_{i=1}^{n}\left(\hat{y_{i}}-y_{i}\right)$$
where *n* is the number of samples, $y_{i}$ is the true value of SOC content, $\bar{y}$ is the average value of SOC content, and $\hat{y}$ is the predicted value of SOC content. A higher R^2^ value indicates a more stable model and a better fitting performance. A lower RMSE value reflects greater accuracy and consistency in the model’s predictive capability.

### 2.6. Trend of Change

To evaluate the spatio-temporal variation trends of SOC in the study area from 2020 to 2024, this paper applied the Slope analysis method to examine the spatial patterns of SOC distribution. The principle involves calculating the rate of change in SOC for each pixel between 2020 and 2024, thereby reflecting the temporal dynamics of regional SOC. Furthermore, when investigating the primary factors influencing changes in SOC, it is essential to perform a trend analysis of ecological environment indicators that exhibit annual variations.(7)$$Slope=\frac{n\sum_{i=1}^{n}i\times y_{i}-\sum_{i=1}^{n}i\sum_{i=1}^{n}y_{i}}{n\sum_{i=1}^{n}i^{2}-{\left(\sum_{i=1}^{n}i\right)}^{2}}$$
where *Slope* is the rate of change. When *Slope* > 0, it indicates that the target is on an upward trend; when it is the converse, it shows a downward trend. *n* is the time span under consideration, and $y_{i}$ is the value of the target variable in the year *i*.

## 3. Results

### 3.1. Model Performance Evaluation

#### 3.1.1. Selection of Key Spectral Bands

Following image preprocessing, seven mathematical transformations were applied to the seven spectral bands of Landsat-9 and the four spectral bands of GF-1. The resulting spectral characteristics are presented in Figure 3. The higher the SOC content was, the lower the band reflectance was. Band selection was performed using the RF-RFECV model. Based on the 95% threshold criterion, a total of 15 band subsets that exerted the most significant influence on model performance were identified (Figure 4). The cross-validation process produced an R^2^ of 0.7693 and an RMSE of 0.6822%. Independent testing set evaluation demonstrated an R^2^ of 0.6001 with an associated RMSE of 0.6221%. Therefore, 15 mathematical transformation bands, which demonstrated significant importance in the inversion of SOC content, were ultimately selected (Table 4).

#### 3.1.2. SOC Estimation Models’ Performance

Among the models of MLR, PLSR, RF, and XGBoost, the XGBoost model had the highest accuracy on both the training set and the testing set (Table 5). The R^2^ of the training set was 0.91, and the RMSE was 0.38%. The R^2^ of cross-validation was 0.8452 and the RMSE was 0.4508%. The R^2^ of the testing set was 0.79, and the RMSE was 0.61%. This is because the relationship between SOC and remote sensing spectral reflectance is highly complex and exhibits strong nonlinearity. Linear models, such as PLSR and MLR, are limited in their ability to fully capture such complexity. The spatial variations in land use patterns and the physical and chemical properties of soil within the Tongken River Basin may result in notable differences in the relationship between SOC content and spectral characteristics across different regions. Although RF is capable of handling data heterogeneity, it is less effective than XGBoost in accurately capturing local feature patterns. This finding is consistent with the research results of Ye et al. [44]. After hyperparameter tuning and cross-validation, the difference in R^2^ between the training set and the testing set of the XGBoost model was less than 0.15 (ΔR^2^ = 0.12), which fell below the tolerance threshold of 0.15 established by Rossel et al. in their study on soil spectral modeling [45]. Meanwhile, the model’s R^2^ values for the inversion of SOC in 2022 and 2024 both exceeded 0.75. Therefore, the XGBoost model exhibited no signs of overfitting in predicting SOC and demonstrated strong cross-temporal and cross-spatial reproducibility, as well as effective transfer generalization performance.

### 3.2. Spatial and Temporal Distribution of SOC

Based on band selection and optimal model construction, this paper predicted the annual SOC status of the Tongken River Basin at a spatial resolution of 16 m from 2020 to 2024. As presented in Table 6, the average SOC content initially increased and subsequently decreased from 2020 to 2024. However, the SOC content in the study area showed an upward trend, with the average value increasing by 22.78%. Spatially, the highest SOC content was observed in the northeastern part of the study area (Figure 5). The SOC content in the study area is higher in paddy fields located along riverbanks compared to that in adjacent dryland areas. This is primarily attributed to the prolonged waterlogging in paddy fields, which fosters an anaerobic environment. This condition markedly reduces the rate at which microorganisms decompose organic matter, thereby facilitating the accumulation of organic carbon [46].

### 3.3. Trends in SOC Changes

The interannual variation in SOC in the Tongken River Basin from 2020 to 2024 was quantified using the Slope trend analysis method, with the Slope value directly representing the absolute magnitude of annual SOC change. The overall trend in the study area indicates an increase, particularly in the hilly region located in the northeast direction, which has experienced a comparatively significant rise in organic carbon levels (Figure 6). The primary reason is that vegetation coverage in this region has increased in recent years, resulting in greater accumulation of litter and root exudates, thereby enhancing the level of SOC. Since 2015, Heilongjiang Province has been promoting the “Longjiang Model,” which integrates straw returning to the field with deep loosening and rotational tillage. In pilot areas such as Wangkui County and Qing’an County, a slight upward trend in SOC has been observed [47]. The regions characterized by a decline in SOC are predominantly located along the gullies adjacent to riverbanks on both sides of the rivers. The area was reduced by 3835.29 km^2^, which accounts for 36.61% of the total area. This phenomenon is primarily attributed to the steep slopes of the gullies on either side of the river, which facilitate rapid runoff generated by rainfall or snowmelt. This swift flow easily erodes the surface soil that is rich in organic matter—particularly in black soil regions—leading to the loss of SOC in conjunction with sediment transport.

### 3.4. Factors Influencing Changes in SOC

Through correlation analysis, the primary factors influencing the development of SOC at the small-watershed scale were identified, and potential multicollinearity among these factors was examined using partial correlation analysis. Within small watersheds, the primary factor influencing the accumulation and variation in SOC is the distribution of soil-forming parent, which exhibits the highest correlation coefficient of 0.38 (Figure 7, Table 7). Furthermore, after excluding the influence of other environmental variables, the association between the two factors becomes significantly stronger (partial correlation coefficient = 0.57). Therefore, the parent material of the soil serves as a primary and independent controlling factor that significantly influences the spatial distribution pattern of SOC in the region. Land use exhibits the second strongest correlation. However, its partial correlation coefficient is 0.07 with statistical insignificance (*p* > 0.05). This indicates that the association may be overestimated, and land use itself does not function as an independent controlling factor for SOC development in this region. Soil type and elevation also exert some influence on the variation in SOC. However, the correlation coefficients are relatively low. Variations in temperature and precipitation exhibit a relatively minor influence on changes in SOC in the Tongken River Basin, with a correlation coefficient below 0.01.

## 4. Discussion

### 4.1. Comparisons of Feature Band Selection Methods

Remote sensing spectral data (104 bands) exhibit high dimensionality, nonlinear inter-band correlations, and multicollinearity characteristics. Comparative studies based on XGBoost demonstrated that RF-RFECV achieved optimal performance in screening spectral-sensitive bands for SOC estimation (Table 8). Compared to conventional techniques, this approach effectively overcomes the loss of spectral physical significance inherent in PCA dimensionality reduction while resolving limitations of linear methods like Least Absolute Shrinkage and Selection Operator (LASSO) in handling complex nonlinear band couplings. Its core advantage lies in integrating the stability of random forests with the precision of recursive elimination, enabling both retention of physical meaning in key spectral bands and refined feature selection within high-dimensional complex data.

### 4.2. Comparisons of Prediction Performance Between Multi-Source Data Fusion and Single-Sensor-Image Model

By integrating data and applying machine learning models, the spectral sensitivity of Landsat and the texture details of GF-1 can be effectively combined, thereby significantly improving the accuracy of SOC inversion and enabling multi-scale mapping across field and regional levels. However, if both satellite bands and their mathematical transformations are incorporated into model construction, although this approach may theoretically provide more abundant information, an excessive number of features can easily lead to overfitting [48]. This results in strong performance on training sets but poor generalization capability. Taking the XGBoost model as an example, compared to using only a single satellite band, the R^2^ of the multi-source data fusion training set increased by 10% to 12% (Table 9). When the SOC model was developed for all spectral bands, although the R^2^ and RMSE values of the training set were satisfactory, the evaluation metrics of the test set exhibited notable discrepancies compared to those of the training set. This indicates that the model exhibited a clear overfitting behavior and was therefore unable to accurately predict SOC content [49]. Additionally, the model was validated using measured SOC data from 2022 and 2024 (Table 5). The validation results demonstrated that the proposed optimal model exhibited strong temporal transferability.

The manuscript successfully estimated the SOC content in the Tongken River Basin through the integration of GF-1 and Landsat-9 remote sensing data, combined with the application of the XGBoost machine learning model. And the model should have good scalability in black soil areas similar to the Tongken River Basin. However, the model’s temporal and spatial generalizability is significantly limited by various factors, including interannual climate variability, shifts in land use and management practices, soil types, the temporal sensitivity of the model, and discrepancies in sampling periods. The validation results obtained from the current sample should be regarded as preliminary references only. The robustness of the findings requires further verification through independent datasets of larger scales, encompassing a wider range of environmental conditions and extended temporal coverage. Subsequently, we will incorporate environmental covariates into the model framework and establish a long-term soil monitoring network. This initiative will systematically accumulate multi-year empirical datasets to enable continuous validation and dynamic model updating, thereby significantly enhancing the model’s temporal robustness and cross-regional transferability.

### 4.3. Comparisons of Slope Analysis and Difference Analysis for Changes in SOC

The SOC in the Tongken River Basin exhibited a fluctuating yet overall upward trend characterized by a “rising–falling–rising” pattern from 2020 to 2024, reflecting the combined influences of agricultural practice transformation, policy implementation, and climatic disturbances. The implementation of the “Longjiang Model” and “Corn–Soybean” crop rotation system in 2021 resulted in a substantial increase in SOC levels between 2020 and 2021 [50]. Following the policy adjustments implemented in 2022, there was a decline in the organic carbon content from 2022 to 2023. The rebound in SOC levels in 2024 can be attributed to the implementation of newly applied technologies such as manure resource utilization and biological soil crusts, along with the enforcement of balanced grazing management.

Therefore, the accumulation or depletion of SOC is influenced by multiple factors and may exhibit phased fluctuations or undergo critical transitions [51]. Using only the difference between the initial and final years may fail to capture critical intermediate variations and turning points in the quantitative assessment of SOC dynamics [52]. By comparing Figure 6 and Figure 8, and taking Point A and Point B as examples, a differential analysis revealed that the organic carbon content at Point A has increased significantly. However, it was found that the SOC content at Point B has shown a greater increasing trend over the past five years through Slope analysis (Table 10). If only the difference is evaluated, the actual potential for an increase in carbon sequestration at Point B may be underestimated.

However, it is essential to acknowledge the inherent limitations associated with optical remote sensing, particularly the issue of data loss resulting from cloud cover interference. The non-random nature of cloud cover may introduce potential biases in the estimation of SOC during specific periods or within certain regions, thereby increasing the noise level in Slope detection. Therefore, future research could integrate microwave remote sensing data or a denser ground-based observation network to further constrain and validate SOC trends retrieved from remote sensing inversion.

### 4.4. Analysis of Key Factors Influencing SOC

Most studies indicate that climate is the primary factor influencing the development of SOC [53,54]. However, the variations in temperature and precipitation exhibited a relatively weak correlation with the changes in SOC in the Tongken River Basin from 2020 to 2024 because the spatial variability of climate change is relatively minor within the scale of small watersheds. Furthermore, the effects of climate change on SOC typically require a long time to become evident. As a result, within a five-year timeframe, these effects may remain comparatively weak [55]. However, the geological background (parent material composition) can directly influence the input of SOC, as well as the decomposition environment and the physicochemical protection mechanisms, thereby serving as key determinant of SOC dynamics. This conclusion aligns with the findings of Wiesmeier et al., who demonstrated that at the small-watershed scale, parent material accounts for the majority of the variability in SOC, whereas climatic factors contribute less than 10% [56].

Although a correlation exists between land use patterns and changes in SOC, the actual impact is not independent, as indicated by the partial correlation coefficient (Table 7). Based on the interaction effects analysis using SHAP values, it can be further revealed that under the same land use type premise, the regulatory effects of different parent materials on SOC vary significantly (Figure 9). Among these, in grassland and forest land use types, the variability in SHAP values induced by differences in parent material is most pronounced, indicating that land use practices exert influence by modulating the intensity of parent material effects. This regulatory role is particularly critical in grassland and forest ecosystems. In contrast, other land use types exhibit considerably less variability induced by parent material, indicating that management practices or human disturbances may mask or diminish the inherent influence of parent material.

## 5. Conclusions

This paper proposes a space–ground collaborative remote sensing framework that integrates data from Landsat-9 and GF-1 satellites to develop an inversion and dynamic monitoring model for black soil SOC. The main conclusions are as follows:(1)The integration of Landsat-9 and GF-1 multi-source remote sensing data effectively addressed the limitations associated with single-source data, and the R^2^ value increased by 10%. The XGBoost model demonstrated superior performance in estimating SOC content in black soil (R^2^ = 0.9130; RMSE = 0.3834%).(2)Based on the optimal model, an assessment of SOC content for the period 2020–2024 was conducted. The average value of SOC in the Tongken River Basin exhibited an initial increase followed by a decrease. From a spatial distribution perspective, the SOC content in the northeastern hilly region exhibited a marked increasing trend.(3)In small watersheds, the primary factors influencing the accumulation and variation in SOC are the soil parent material.

This method can serve as a case reference for large-scale and high-frequency SOC monitoring in global black soil regions.

## Figures and Tables

**Figure 1 sensors-25-05442-f001:**
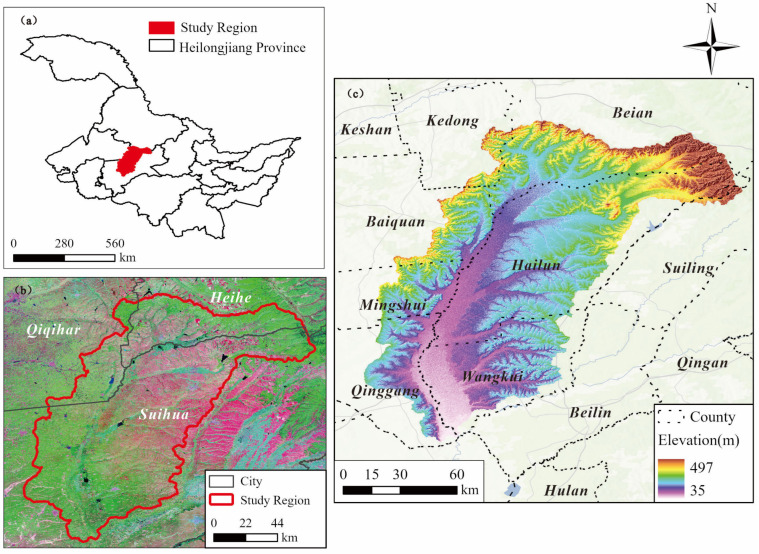
Location map of the Tongken River Basin. (**a**) The location of Heilongjiang, China. (**b**) Remote sensing image of the Tongken River Basin. (**c**) Elevation distribution of the Tongken River Basin.

**Figure 2 sensors-25-05442-f002:**
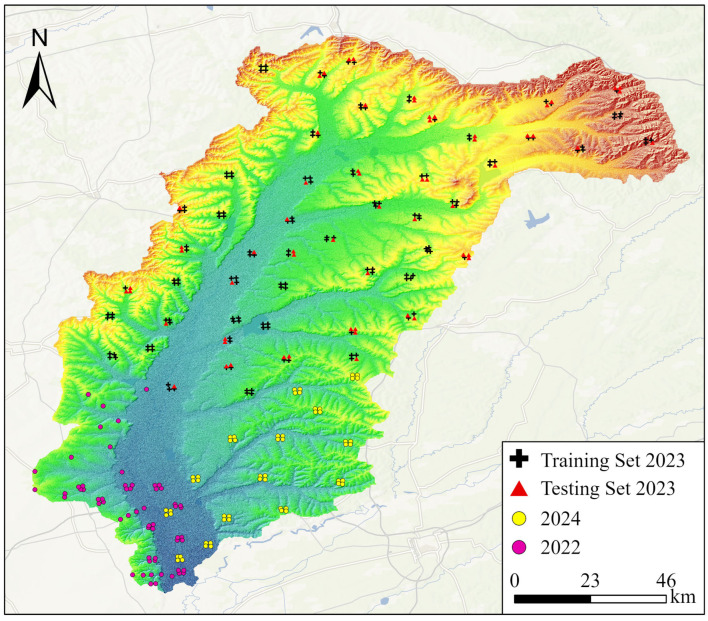
The distribution of sampling points in the Tongken River Basin.

**Figure 3 sensors-25-05442-f003:**
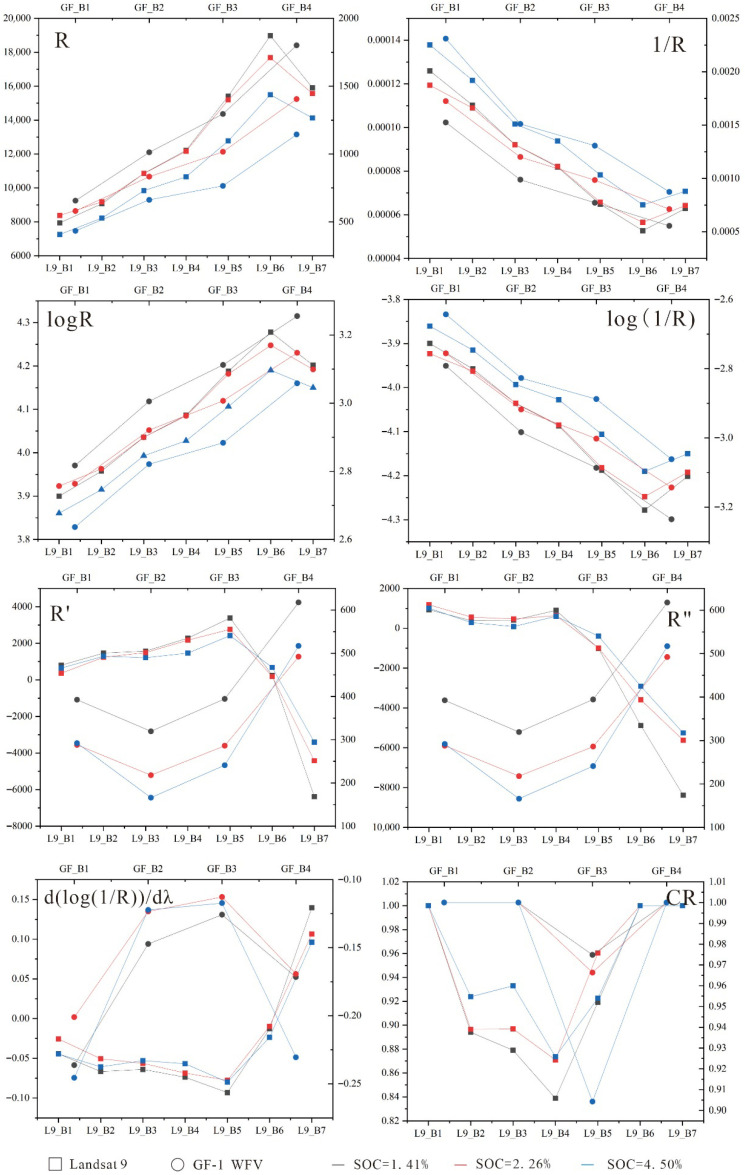
Spectral transformation curve.

**Figure 4 sensors-25-05442-f004:**
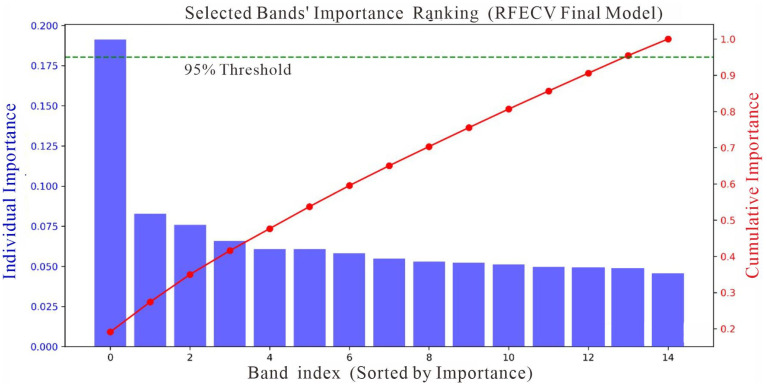
RF-RFECV importance threshold curve.

**Figure 6 sensors-25-05442-f006:**
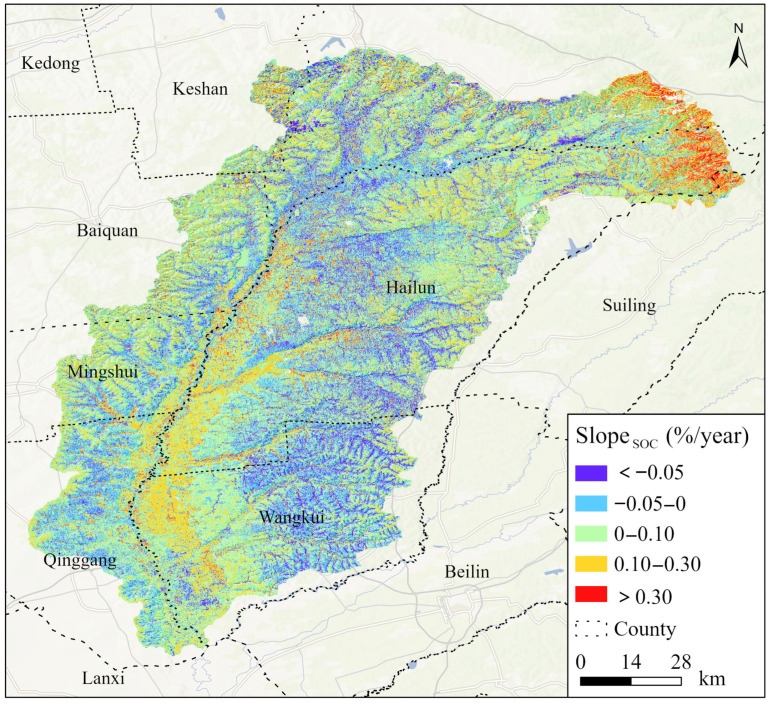
Trend in SOC changes from 2020 to 2024.

**Figure 7 sensors-25-05442-f007:**
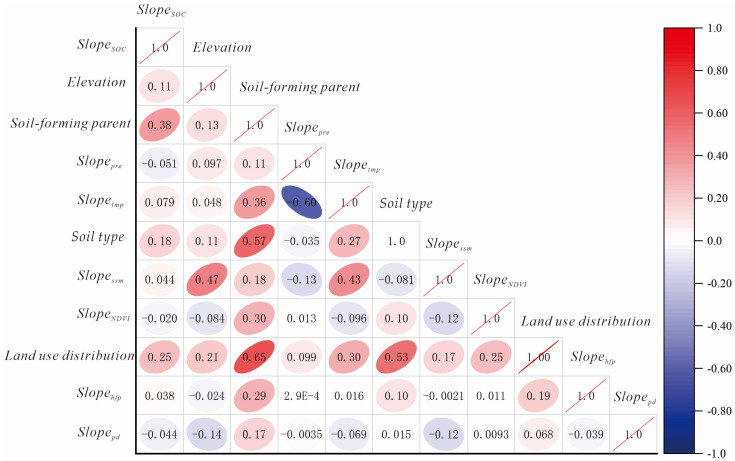
Correlation matrix of factors influencing SOC change.

**Figure 5 sensors-25-05442-f005:**
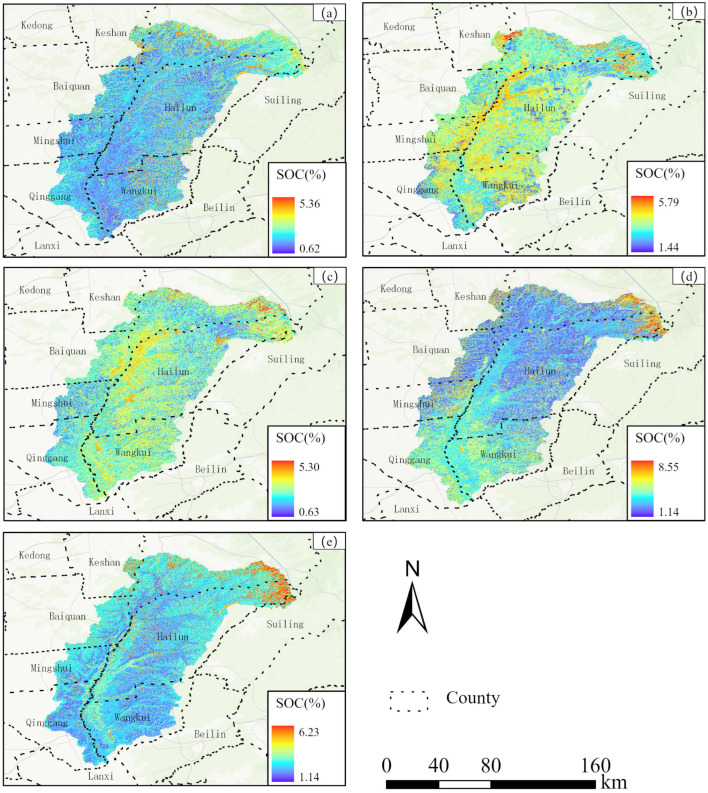
The spatial distribution of SOC content in the Tongken River Basin ((**a**). 2020, (**b**). 2021, (**c**). 2022, (**d**). 2023, (**e**). 2024).

**Figure 8 sensors-25-05442-f008:**
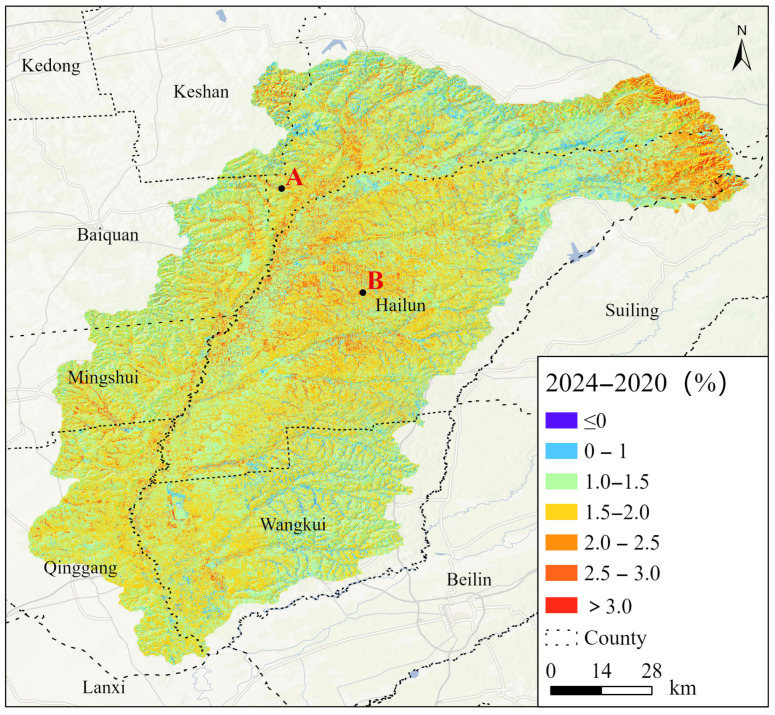
Spatial distribution of the increase in SOC from 2020 to 2024 (A, B were the points of comparison.).

**Figure 9 sensors-25-05442-f009:**
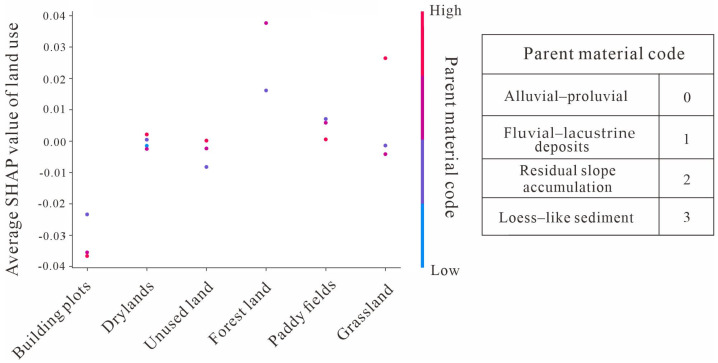
SHAP interaction effects between land use and parent material.

**Table 1 sensors-25-05442-t001:** The acquisition time of satellite images.

Year	Landsat-9 OLI-/TIRS L2	GF-1 WFV
2020	2020.04.10	2020.04.15, 2020.05.06
2021	2021.03.28	2021.04.07
2022	2022.03.23, 2022.03.21	2022.04.02, 2022.04.15
2023	2023.04.27, 2023.04.04	2023.05.17, 2023.05.18
2024	2024.04.29, 2024.05.07	2024.05.09

**Table 2 sensors-25-05442-t002:** Statistical information on organic carbon content in soil samples.

Year		Number	Maximum (%)	Minimum (%)	Average (%)
2022		60	3.80	0.65	1.98
2023	Training set	143	8.42	1.4	3.26
	Testing set	61	8.20	1.66	3.04
2024		56	3.03	1.11	1.9

**Table 3 sensors-25-05442-t003:** Indicators and data sources.

	Indicators	Data Content	Resolution (m)	Format	Source
Topography and geology	Elevation	Elevation	30	GeoTiff	Geospatial Data Cloud (https://www.gscloud.cn/home accessed on 20 March 2024)
	Distribution of soil-forming parent	Soil-forming parent	——	Shapefile	National Earth System Science Data Center (http://soil.geodata.cn/ accessed on 20 February 2024)
Meteorological factors	Mean annual temperature change Slope $\left({Slope}_{tmp}\right)$	Mean annual temperature	1000	GeoTiff	National Tibetan Plateau Data Center (https://data.tpdc.ac.cn/home accessed on 14 February 2024)
	Mean annual precipitation change Slope $\left({Slope}_{pre}\right)$	Mean annual rainfall	1000	GeoTiff	
Soil characteristics	Surface soil moisture change Slope $\left({Slope}_{ssm}\right)$	Surface soil moisture	1000	GeoTiff	
	Distribution of soil type	Soil type	——	Shapefile	National Earth System Science Data Center (http://soil.geodata.cn/ accessed on 22 February 2024)
Ecological environment	Vegetation index change Slope $\left({Slope}_{NDVI}\right)$	Normalized difference vegetation index	250	GeoTiff	Resource and Environmental Science Data Platform (https://www.resdc.cn/ accessed on 5 March 2024)
Human activities	Land use distribution	Land use data	——	Shapefile	The data center of the Institute of Resources and Environment, Chinese Academy of Sciences (https://www.resdc.cn/ accessed on 20 May 2024)
	Annual terrestrial Human Footprint change Slope $\left({Slope}_{hfp}\right)$	Annual terrestrial Human Footprint	1000	GeoTiff	Scientific Data [31] (https://www.nature.com/sdata/ accessed on 22 May 2024)
	Population density change Slope $\left({Slope}_{pd}\right)$	LandScan Global	1000	GeoTiff	Oak Ridge National Laboratory (https://landscan.ornl.gov/about accessed on 23 May 2024))

**Table 4 sensors-25-05442-t004:** The feature bands selected by RF-RFECV.

Mathematical Transformation	GF-1	Landsat-9
R	B1, B2, B4	B1, B4
1/R	NULL	B1
logR	NULL	B1, B2
log(1/R)	B2	B7
R′	NULL	NULL
R″	NULL	B4
d(log(1/R))/dλ	B2, B3	B6, B7
CR	NULL	NULL

**Table 5 sensors-25-05442-t005:** Prediction accuracy of different models for SOC.

Model	Training Set				Testing Set			
	R^2^	RMSE (%)	MAE (%)	Bias (%)	R^2^	RMSE (%)	MAE (%)	Bias (%)
MLR	0.3603	1.0042	0.7181	0.0001	0.2238	1.2937	0.7333	−0.0950
PLSR	0.4356	0.9985	0.6513	0.0001	0.3411	1.0552	0.7841	−0.0151
RF	0.7206	0.6271	0.2669	0.1449	0.5951	0.6312	0.4574	0.2607
XGBoost	0.9130	0.3834	0.1087	−0.0005	0.7869	0.6134	0.3233	0.2312
XGBoost (2022)	R^2^ = 0.8253		RMSE = 0.2868%		MAE = 0.2465%		Bias = 0.2069%	
XGBoost (2024)	R^2^ = 0.7702		RMSE = 0.4807%		MAE = 0.1118%		Bias = 0.1003%	

**Table 6 sensors-25-05442-t006:** Statistics of SOC from 2020 to 2024.

Time	SOC (%)		
	Maximum	Minimum	Average
2020	5.3641	0.6247	3.3776
2021	5.7928	1.4402	4.4945
2022	5.3029	0.6303	4.0124
2023	8.5500	1.3923	3.3193
2024	6.2263	1.1393	4.1469

**Table 7 sensors-25-05442-t007:** Correlation coefficients and partial correlation coefficients.

	${Slope}_{SOC}$	Elevation	Soil-Forming Parent	${Slope}_{pre}$	${Slope}_{tmp}$	Soil Type	${Slope}_{ssm}$	${Slope}_{NDVI}$	Land Use Distribution	${Slope}_{hpf}$	${Slope}_{pd}$
${Slope}_{SOC}$	1	0.11 **	0.38 **	−0.051 **	0.079 **	0.18 **	0.044 **	−0.02	0.25 **	0.038 **	−0.044 **
Elevation	0.11 **	1	0.13 **	0.097	0.048 **	0.11 **	0.47 **	−0.084 **	0.21 **	−0.024 *	−0.14 **
Soil-forming parent	0.38 **	0.13 **	1	0.11 **	0.36 **	0.57 **	0.18 **	0.30 **	0.65 **	0.29 **	0.17 **
${Slope}_{pre}$	−0.051 **	0.097 **	0.11 **	1	−0.60 **	−0.035 *	−0.13 **	0.013	0.099 **	2.90 × 10^−4^	−0.0035
${Slope}_{tmp}$	0.079 **	0.048 **	0.36 **	−0.60 **	1	0.27 **	0.43 **	−0.096 **	0.30 **	0.016	−0.069 **
Soil type	0.18 **	0.11 **	0.57 **	−0.035 *	0.27 **	1	−0.08 **	0.10 **	0.53 **	0.10 **	0.015
${Slope}_{ssm}$	0.044 **	0.47 **	0.18 **	−0.13 **	0.43 **	−0.08 **	1	−0.12 **	0.17 **	−0.0021	−0.12 **
${Slope}_{NDVI}$	−0.020	−0.084 **	0.30 **	0.013	−0.096 **	0.10 **	−0.12 **	1	0.25 **	0.011	0.0093
Land use distribution	0.25 **	0.21 **	0.65 **	0.099 **	0.30 **	0.53 **	0.17 **	0.25 **	1	0.19 **	0.068 **
${Slope}_{hpf}$	0.038 **	−0.024 *	0.29 **	2.9 × 10^−4^	0.016	0.10 **	−0.0021	0.011	0.19 **	1	−0.039 **
${Slope}_{pd}$	−0.044 **	−0.14 **	0.17 **	−0.0035	−0.069 **	0.015	−0.12 **	0.0093	0.068 **	−0.039 **	1
Partial correlation coefficient	1	0.184	0.57	−0.026	0.051	−0.063	−0.043	0.011	0.07	0.005	0.008
Significance (P)	0	0	0	0	0	0	0	0.142	0.197	0.470	0.272

* The correlation is statistically significant at the 0.01 significance level. ** The correlation is statistically significant at the 0.05 significance level.

**Table 8 sensors-25-05442-t008:** Accuracy comparison of feature band selection models.

		PCA	LASSO
Number		6	10
Training set	R^2^	0.5894	0.7037
	RMSE(%)	0.6156	0.6430
	MAE(%)	0.4783	0.3113
	Bias(%)	−0.3593	0.2113
Testing set	R^2^	0.3066	0.6610
	RMSE(%)	1.9708	0.6889
	MAE(%)	0.6409	0.3131
	Bias(%)	−0.4972	0.3131

**Table 9 sensors-25-05442-t009:** Comparison of model construction accuracy.

Satellite Type	All Bands				RF-RFECV Spectral Band Extraction			
	Training Set		Testing Set		Training Set		Testing Set	
	R^2^	RMSR (%)	R^2^	RMSR (%)	R^2^	RMSR (%)	R^2^	RMSR (%)
Landsat	0.8129	0.4814	0.4881	0.9750	0.8297	0.4452	0.5032	0.6009
GF-1	0.8288	0.4641	0.4244	0.9908	0.8300	0.4046	0.5018	0.6028
Landsat-9 and GF-1	0.9123	0.3828	0.5329	0.6048	0.9130	0.3834	0.7869	0.6134

**Table 10 sensors-25-05442-t010:** Annual statistics and changes in SOC at locations A and B (%).

	2020	2021	2022	2023	2024	Slope	Difference
A	3.3712	5.0561	4.1556	3.1128	5.2314	0.1810	2.9531
B	3.2902	4.5168	3.8194	3.4214	5.3673	0.3199	2.8868

## Data Availability

The raw data supporting the conclusions of this article will be made available by the authors on request.

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
