# Peer review of "Intelligent Monitoring and Trend Analysis of Surface Soil Organic Carbon in the Black Soil Region Using Multi-Satellite and Field Sampling: A Case Study from Northeast China"

_sensors, 2025, doi:10.3390/s25175442_

Round 1
Reviewer 1 Report
Comments and Suggestions for Authors
- According to which soil classification are soils named - Black Soil? Give the name of the soil by WRB (world reference base soil).
- Provide a morphological description of the soil. What is the thickness of the humus-accumulative horizon?
- Line 165. Please report sampling design.
- Line 236. Did you tune a RF model?
- Software used should report.
- What about the cross-validation method? I don't see it, but it's very important.
- Table 6. and Figure 5. What is the reason for the strong change in SOC content over the years? For example, from 2020 to 2021 there was an increase in content by 1.1169 %, then a decrease in 2023 and an increase again in 2024 (average data) throughout the studied region? Please give a more detailed explanation.
Author Response
Dear Reviewer:
Thank you for your comments concerning our manuscript. These comments are very helpful for revising and improving our paper. We have studied the comments carefully and made corrections which we hope meet with approval. The revised parts in the manuscript were highlighted in yellow.
Question 1:
According to which soil classification are soils named - Black Soil? Give the name of the soil by WRB (world reference base soil).
Answer:
Northeast China is one of the world's four major black soil regions and a key global grain-producing area. To strengthen the protection of black soil, the Chinese government has enacted the 'Law of the People's Republic of China on the Protection of Black Soil.' According to relevant legal provisions, black soil is defined as soil with a black or dark black humus surface layer, characterized by good structure and high fertility. We adhere to this definition. In the World Reference Base classification, the majority of black soils would correspond to chernozems, kastanozems and phaeozems. It is not in conflict with the WRB but places greater emphasis on the perspective of benefiting grain production. To ensure a clear and comprehensive description of black soil, a dedicated introduction has been included in the introduction (Line 40-46). At the same time, relevant references were supplemented (Line 583, 585).
Line40-46: Northeast China is one of the world's four major black soil regions and a key global grain-producing area. To strengthen the protection of black soil, the Chinese government has enacted the 'Law of the People's Republic of China on the Protection of Black Soil.' According to relevant legal provisions, black soil is defined as soil with a black or dark black humus surface layer, characterized by good structure and high fertility. In the World Reference Base classification, the majority of black soils would correspond to chernozems, kastanozems and phaeozems[1,2].
Line 582: [1] Jian, W.; Qi, X.; Yuxin, T.; Zhenzhong, W.; Baoguo, L. Establish governance systems for securing black soils in China based on a new law. Soil Secur. 2023, 13, 100112. https://doi.org/ 10.1016/j.soisec.2023.100112.
Line 584: [2] Hag Husein, H.; Bumler, R.; Lucke, B.; Sahwan, W. Black Soils in the Eastern Mediterranean: Genesis and Properties. Geographies 2024, 4, 168-181. https://doi.org/10.3390/geographies4010011.
Question 2:
Provide a morphological description of the soil. What is the thickness of the humus-accumulative horizon?
Answer:
Appropriate introductory content was added to the "2.1. Study Area" section. At the same time, the relevant reference was supplemented:
Line157-158:The thickness of the soil humus layer generally ranges from 20 to 50 cm, and in certain regions, it may extend to 80 to 100 cm [23].
Line 635: [23] Wang, J.; Zuo, L.; Li, Z.; Mu, H.; Zhou, P.; Yang, J.; Zhao, Y.; Qin, K. A detection method of trace metal elements in black soil based on hyperspectral technology: Geological implications. J. Geomech. 2021, 27, 418-429. https://doi.org/10.12090/j.issn.1006-6616.2021.27.03.038
Question 3:
Line 165. Please report sampling design.
Answer:
Considering the representativeness and spatial heterogeneity of soil in the Tongken River Basin, this study employed a stratified random sampling method based on spatial stratification of land use and elevation to establish sample points in 2023. Land use was categorized into drylands, paddy fields, forest land, grassland, water bodies, construction land, and unused land. The elevation was classified into the following elevation ranges: <100 m, 100–200 m, 200–300 m, 300–400 m, and >400 m. Through a hierarchical design approach, the entire basin was subdivided into 66 distinct regions. Except for water bodies, three to four sample points are randomly selected in each area. Following outlier removal, the 2023 dataset comprised 204 validated soil samples, which were partitioned into training set and testing set (7:3) for model development and validation. And the testing set points were prioritized in larger areas to ensure adequate spatial coverage of the validation data.
Additional explanations regarding the sampling design have been provided in "2.2.2 Soil Sample Collection" (Line 202-219) :
Line 202-219: In this manuscript, a multi-source data fusion model for SOC in black soil was developed based on the measured SOC data in 2023. Considering the representative-ness and spatial heterogeneity of soil in the Tongken River Basin, this study employed a stratified random sampling method based on spatial stratification of land use and elevation to establish sample points in 2023. Through a hierarchical design approach, the entire basin was subdivided into 66 distinct regions. Except for water, three to four sampling points were randomly established in each of these regions. Soil samples were collected from the surface layer (0–20 cm). Following natural air-drying, the samples were sieved using a 20-mesh nylon sieve. The SOC content was determined using the potassium dichromate oxidation method with external heating [28]. Following the exclusion of outliers, a total of 204 soil samples were collected in 2023. The sample points were divided into a training set and a testing set at a ratio of 7:3. To ensure adequate spatial coverage of the validation data, the testing set was preferentially distributed across larger areas. To assess the model's capability in capturing recent trends and its short-term extrapolation performance, this study employed 60 soil samples from 2022 and 56 soil samples from 2024 to validate the optimal model's simulated SOC distributions for the corresponding years. The distribution of sampling points is shown in Figure 2 and the statistical information is shown in Table 2.
Question 4:
Line 236. Did you tune a RF model?
Answer:
We conducted fine-tuning of the RF model using Randomized Search. In “2.4.3 RF”, the tuning methods for RF and the associated hyperparameter settings have been added (Line 297-304).
Line 297-303: In this manuscript, hyperparameters optimization of the RF model was conducted using Randomized Search method, with performance evaluation of different hyperparameters combinations carried out through 10-fold cross-validation. Following the optimization through random search, the final set of hyperparameters was deter-mined as follows: the number of trees is 400, the maximum tree depth is 50, the mini-mum number of samples required for a split is 2, the minimum number of samples required at a leaf node is 1, and the proportion of features considered for splitting is 0.7.
Question 5:
Software used should report.
Answer:
Remote sensing image preprocessing and spectral mathematical transformations were performed using ENVI 5.6. The geographical covariates were processed using QGIS 3.1. Spectral band extraction and model development for SOC content were performed using Python programs developed in JetBrains PyCharm 2019 (Community Edition). The use of the software was appropriately supplemented at the relevant section of the manuscript:
Line 198-200: All remote sensing image processing tasks and spectral mathematical transformations were conducted using ENVI 5.6.
Line 231: These operations were conducted using the QGIS 3.1.
Line 252-253: This method was implemented through the development of a Python program using JetBrains PyCharm 2019 (Community Edition).
Line260-261: All methods were implemented using JetBrains PyCharm 2019 (Community Edition).
Question 6:
What about the cross-validation method? I don't see it, but it's very important.
Answer:
In this study, cross-validation is primarily employed for feature band selection using RF-RFECV (Line 250-252, Line 347-349), as well as for hyperparameters tuning of the RF (Line 297-303) and XGBoost models (Line 313-318, Line 360-361). The manuscript has been updated to include detailed explanations of cross-validation and the associated results at the relevant sections.
Line 250-252: And ten-fold cross-validation is employed to ensure that the selected feature subset exhibits optimal predictive performance on unseen data.
Line 347-349: The cross-validation process consistently produces an R² of 0.7693 and an RMSE of 0.6822%. Independent testing set evaluation demonstrates an R² of 0.6001 with an associated RMSE of 0.6221%.
Line 297-303: In this manuscript, hyperparameters optimization of the RF model was conducted using Randomized Search method, with performance evaluation of different hyperparameters combinations carried out through 10-fold cross-validation. Following the optimization through random search, the final set of hyperparameters was determined as follows: the number of trees is 400, the maximum tree depth is 50, the minimum number of samples required for a split is 2, the minimum number of samples required at a leaf node is 1, and the proportion of features considered for splitting is 0.7.
Line 313-318: In this manuscript, the hyperparameters of the XGBoost model were optimized using the Randomized Search method in conjunction with 10-fold cross-validation. The final hyperparameters, determined through random search optimization, are as follows: the number of trees is set to 200, the maximum tree depth is 5, the learning rate is 0.05, the sampling ratio for both samples and features is 0.9, the minimum loss reduction required for node splitting is 0.1, and all regularization coefficients are set to 0.5.
Line 360-361: The R2 of cross-validation is 0.8452 and the RMSE is 0.4508%.
Question 7:
Table 6. and Figure 5. What is the reason for the strong change in SOC content over the years? For example, from 2020 to 2021 there was an increase in content by 1.1169 %, then a decrease in 2023 and an increase again in 2024 (average data) throughout the studied region? Please give a more detailed explanation.
Answer:
The explanation regarding the changes in SOC has been incorporated into the Discussion (Line 486-495), At the same time, relevant reference has been supplemented (Line 699).
Line 486-495: The SOC in the Tongken River Basin exhibited a fluctuating yet overall upward trend characterized by a "rising-falling-rising" pattern from 2020 to 2024, reflecting the combined influences of agricultural practice transformation, policy implementation, and climatic disturbances. The implementation of the “Longjiang Model” and “Corn-Soybean" crop rotation system in 2021 resulted in a substantial increase in SOC levels between 2020 and 2021[50]. Following the policy adjustments implemented in 2022, there was a decline in the organic carbon content from 2022 to 2023. The rebound in SOC levels in 2024 can be attributed to the implementation of newly applied technologies such as manure resource utilization and biological soil crusts, along with the enforcement of balanced grazing management.
Line 699: [50] Zhang, J.; Sun, B.; Zhu, J; Wang, J.; Pan, X.; Gao, T. Black Soil Protection and Utilization Based on Harmonization of Mountain-River-Forest-Farmland-Lake-Grassland-Sandy Land Ecosystems and Strategic Construction of Ecological Barrier. Bull. Chin. Acad. Sci. 2021, 036, 1155-1164. https://doi.org/10.16418/j.issn.1000-3045.20211010002

Reviewer 2 Report
Comments and Suggestions for Authors
Summary. The manuscript entitled "Intelligent Monitoring and Trend Analysis of Surface Soil Organic Carbon in the Black Soil Region Using Multi-satellite and Field Sampling: A case study from Northeast China" presents a machine learning-based approach to estimate and monitor soil organic carbon (SOC) in the Tongken River Basin using synergistic Landsat-9 and GF-1 satellite data. The authors applied and compared several models (MLR, PLSR, RF, XGBoost), concluding that XGBoost performed best (R² = 0.9130, RMSE = 0.3834%). The generated 16 m SOC maps from 2020 to 2024 demonstrate spatial and temporal dynamics of SOC and suggest parent material and land use as primary drivers. The paper is relevant, well-structured, and offers a significant contribution to soil monitoring and sustainable land management research.
General Concept Comments. This manuscript is timely and scientifically sound. It addresses a critical need for high-resolution SOC monitoring. The integration of field sampling with multi-source satellite data and advanced machine learning represents a well-conceived methodology. The clarity of writing, logical flow, and comprehensive data processing approach make this study accessible and replicable. The results have both scientific and applied value for soil health monitoring in agricultural regions.
Scientific Content Evaluation. The study compares multiple machine learning models, providing a solid foundation for selecting the most appropriate approach (XGBoost). It makes an effective case for combining Landsat-9 and GF-1 imagery for improved SOC inversion accuracy. The validation of the model using real data from 2022 and 2024 demonstrates its transferability. The use of slope analysis to capture SOC change trends over time is appropriate and informative.
The hypothesis is not explicitly stated in the Introduction. While the goals are clear, a specific hypothesis would help frame the research more sharply. The rationale for using certain spectral transformations could be better justified, particularly since such transformations may introduce noise. The selection of 2022 and 2024 for validation is not fully explained – why were these years chosen instead of others within the same dataset? There is limited discussion of potential confounding variables such as soil sampling bias or seasonal variability. The results interpretation could be strengthened by including statistical tests for significance in trend or correlation analyses (e.g., p-values or confidence intervals).
Specific Comments
The abstract is informative, but the hypothesis or research question should be explicitly stated. Consider briefly mentioning why the Tongken River Basin is a suitable case study. (Lines 21–33)
The introduction outlines background and significance well, but it would benefit from a concise statement of the hypothesis or central research question at the end. The correlation between SOC and spectral bands is referenced; consider quantifying this correlation from cited studies to strengthen the argument. (Lines 35–115)
A large number of spectral transformations are applied (e.g., logR, 1/R, CR, derivatives), but there is no discussion of potential redundancy or multicollinearity. Please justify how this was managed or mitigated. The rationale for resampling Landsat-9 to 16 m to match GF-1 is reasonable, but this could introduce artifacts. A brief note on resampling quality control would be useful. (Lines 138–163)
It is unclear how sampling locations were chosen. Were they stratified by land use, topography, or other spatial features? This is critical for understanding spatial representativeness. How was the number of samples determined (especially the limited number in 2022 and 2024)? Justify whether this sampling density is adequate for model validation. (Lines 164–177)
Why was XGBoost expected to outperform the other models in this context? Any specific soil-related reasoning? How were hyperparameters optimized for RF and XGBoost? This should be mentioned briefly for reproducibility. (Lines 205–255)
The validation results for 2022 and 2024 are promising, but more explanation is needed on why these years were selected. Were they extreme or representative years? (Lines 295–299)
The slope classification legend is missing or unclear. Please add or improve the visual scale to make interpretation easier. Include a brief explanation of how slope values relate to absolute changes in SOC (i.e., %/year or units/year). (Figure 6)
The correlation analysis identifies parent material as the dominant factor (R = 0.38). This is informative, but suggest including a table with all correlation coefficients for transparency. Consider discussing whether any multivariate or partial correlation analysis was performed to disentangle co-varying effects. (Lines 336–344)
The conclusions are well-aligned with the findings. It would be valuable to mention the scalability of the model to other regions or soils, and potential integration with operational monitoring systems. (Lines 402–420)
Additional Questions for the Authors
How was the potential temporal mismatch between satellite image acquisition and soil sampling dates addressed in your modeling and validation process?
Can you clarify whether the model performance varied significantly across different land use categories (e.g., paddy fields vs. drylands)?
Author Response
Dear Reviewer:
Thank you for your comments concerning our manuscript. These comments are very helpful for revising and improving our paper. We have studied the comments carefully and made corrections which we hope meet with approval. The revised parts in the manuscript were highlighted in yellow.
Question 1:
The abstract is informative, but the hypothesis or research question should be explicitly stated. Consider briefly mentioning why the Tongken River Basin is a suitable case study. (Lines 21–33)
Answer:
Thank you for your valuable guidance. The Tongken River Basin combines the topographic features, soil properties, and agricultural development and utilization patterns typical of black soil regions. Its soil erosion processes and black soil degradation issues collectively reflect the prevalent ecological challenges faced by the Northeast Black Soil Zone. Therefore, we selected this river basin as the study area. We have accordingly revised the abstract section based on your suggestions.
Line 22-36: The black soil region of Northeast China is a critical global grain production base. The dynamic variations in soil organic carbon (SOC) are directly linked to the regional food security. To accurately monitor SOC content and evaluate the potential of integrating Landsat-9 and GF-1 satellite data for SOC inversion, we developed a machine learning framework that combines data from both satellite sources to model SOC. Using the typical black soil region of Northeast China in the Tongken River Basin as the study area, we compared the MLR, PLSR, RF, and XGBoost algorithms. And XGBoost demonstrated the highest performance (R² = 0.9130, RMSE = 0.3834%). Based on the optimal model, SOC in the study area was projected from 2020 to 2024. The multi-year average SOC exhibited an initial increase followed by a subsequent de-cline, with an overall increase of 22.78%. Spearman correlation analysis identified parent material as the dominant factor controlling SOC variation at the watershed scale (correlation coefficient=0.38) while also modulating the influence of land use types on SOC dynamics. The "Space-ground" multi-source collaborative inversion framework developed in this study offers a high-precision technical approach for the monitoring of SOC in black soil regions
Question 2:
- The introduction outlines background and significance well, but it would benefit from a concise statement of the hypothesis or central research question at the end.
Answer:
Thank you sincerely for your valuable suggestions. The central research question of the manuscript was explicitly stated at the conclusion of the introduction section(Line130-134):
Line 130-134:Therefore, to investigate the feasibility and potential of integrating Landsat and GF satellite data for SOC retrieval, this study aims to address the following key re-search questions: Can the integration of Landsat-9 and GF-1 satellite data significantly enhance the accuracy of remote sensing inversion for SOC content in black soil regions? Which machine learning model performs the best?
- The correlation between SOC and spectral bands is referenced; consider quantifying this correlation from cited studies to strengthen the argument. (Lines 35–115).
We examined the relevant literature and quantified the correlation between SOC and spectral bands in the introduction(Line 63-74).
Line 63-74: Saha et al. carried out a study utilizing Hyperion hyperspectral satellite data collected from regions in western India. The results of the correlation analysis indicate that the spectral reflectance of Hyperion band 57, corresponding to a wavelength of 1033.88 nm, exhibits the highest correlation with SOC content, with a correlation coefficient of -0.86[5]. Castaldi et al. conducted a systematic evaluation of the sensitivity and predictive capacity of multispectral satellite sensors in relation to SOC. Their findings revealed that the SWIR and the red band demonstrated the strongest negative correlations with SOC (correlation coefficient<-0.6) [6]. Gholizadeh et al. systematically elaborated on the spectral response mechanism of SOC in their review, proposing that distinct absorption bands exist for the C=O bond in the range of 2100–2300 nm and for the C=H bond in the range of 1650–1750 nm, which represent optimal wavelength ranges for SOC inversion [7].
Relevant references:
Line 590: [5] Saha, S.K.; Tiwari, S.K.; Kumar, S. Integrated Use of Hyperspectral Remote Sensing and Geostatistics in Spatial Prediction of Soil Organic Carbon Content. J. Indian Soc. Remote Sens. 2022, 1-13. https://doi.org/10.1007/s12524-021-01459-7.
Line 592: [6] Castaldi, F.; Palombo, A.; Santini, F.; Pascucci, S.; Pignatti, S.; Casa, R. Evaluation of the potential of the current and forthcoming multispectral and hyperspectral imagers to estimate soil texture and organic carbon. Remote Sens. Environ. 2016, 179, 54-65. https://doi.org/10.1016/j.rse.2016.03.025
Line 595: [7] Gholizadeh, A.; Saberioon, M.; Ben-Dor, E.; Borůvka, L. Monitoring of selected soil contaminants using proximal and remote sensing techniques: Background, state-of-the-art and future perspectives. Crit. Rev. Environ. Sci. Technol. 2018,.48, 243-278, https://doi.org/10.1080/10643389.2018.1447717.
Question 3:
- A large number of spectral transformations are applied (e.g., logR, 1/R, CR, derivatives), but there is no discussion of potential redundancy or multicollinearity. Please justify how this was managed or mitigated.
Answer:
Thank you for your careful evaluation of the methodological rigor of this research. Although various spectral mathematical transformations can enrich the information content, they may also introduce the issue of multicollinearity. Therefore, RF-RFECV is employed for feature selection in this study. This method is capable of effectively processing high-dimensional spectral data with strong inter-variable correlations. The tree structure splitting mechanism in RF does not depend on the assumption of feature independence and inherently exhibits robustness against multicollinearity. By recursively eliminating the least important features, the method ensures that only the most discriminative features are retained within highly correlated feature groups. Ten-fold cross-validation is employed to ensure that the selected feature subset exhibits optimal predictive performance on unseen data.
In “2.3 RF-RFECV Spectral Band Extraction”, we provided a supplementary description of the collinearity issue (Line 245-252). In “3.1.1 Selection of Key Spectral Bands”, we supplemented the results of cross-validation (Line 347-349). PCA and LASSO are commonly employed methods for addressing the issue of multicollinearity among indicators. To emphasize the advantages of RF-RFECV in spectral feature extraction, a comparative analysis incorporating PCA, LASSO, and RF-RFECV was introduced in “4.1. Comparisons of feature band selection methods” of this manuscript (Line 440-449).
Line 245-252: RF-RFECV machine learning method is capable of effectively processing high-dimensional spectral data with strong inter-variable correlations. The tree structure splitting mechanism in RF does not depend on the assumption of feature independence and inherently exhibits robustness against multicollinearity. By recursively eliminating the least important features, the method ensures that only the most dis-criminative features are retained within highly correlated feature groups. And ten-fold cross-validation is employed to ensure that the selected feature subset exhibits optimal predictive performance on unseen data.
Line 347-349: The cross-validation process produces an R² of 0.7693 and an RMSE of 0.6822%. Independent testing set evaluation demonstrates an R² of 0.6001 with an associated RMSE of 0.6221%.
Line 440-449: Remote sensing spectral data (104 bands) exhibit high dimensionality, nonlinear inter-band correlations, and multicollinearity characteristics. Comparative studies based on XGBoost demonstrate that RF-RFECV achieved optimal performance in screening spectral-sensitive bands for SOC estimation (Table 8). Compared to conventional techniques, this approach effectively overcomes the loss of spectral physical significance inherent in PCA dimensionality reduction while resolving limitations of linear methods like Least Absolute Shrinkage and Selection Operator (LASSO) in handling complex nonlinear band couplings. Its core advantage lies in integrating the stability of random forests with the precision of recursive elimination, enabling both retention of physical meaning in key spectral bands and refined feature selection within high-dimensional complex data.
Table 8. Accuracy comparison of feature band selection models
|
PCA |
LASSO |
|
Number 0.8288 |
6 |
10 |
|
Training set |
R2 |
0.5894 |
0.7037 |
RMSE(%) |
0.6156 |
0.6430 |
|
MAE(%) |
0.4783 |
0.3113 |
|
Bias(%) |
-0.3593 |
0.2113 |
|
Testing set |
R2 |
0.3066 |
0.6610 |
RMSE(%) |
1.9708 |
0.6889 |
|
MAE(%) |
0.6409 |
0.3131 |
|
Bias(%) |
-0.4972 |
0.3131 |
- The rationale for resampling Landsat-9 to 16 m to match GF-1 is reasonable, but this could introduce artifacts. A brief note on resampling quality control would be useful. (Lines 138–163)
Answer:
In "2.2.1 Satellite Retrieval," we have included detailed descriptions of the resampling and quality control methodologies (Line 180-189).
Line 180-189: To achieve the fusion analysis of multi-source remote sensing imagery, spatial resampling is performed on the Landsat-9 images. The spatial resolution of all datasets was uniformly adjusted to 16m using the Bicubic Convolution algorithm. On GF-1 high-resolution images, 20 to 30 permanent ground features that exhibit stable temporal and spatial characteristics, such as road intersections and corners of dam structures, were selected as ground control points. A first-order affine transformation was employed to perform cross-sensor geometric registration between Landsat-9 and GF-1. The registration accuracy control standard specifies that the root mean square error (RMSE) should be less than 0.5 GF-1 multispectral pixel, equivalent to 8m or less
Question 4:
It is unclear how sampling locations were chosen. Were they stratified by land use, topography, or other spatial features? This is critical for understanding spatial representativeness. How was the number of samples determined (especially the limited number in 2022 and 2024)? Justify whether this sampling density is adequate for model validation. (Lines 164–177).
Answer:
Considering the representativeness and spatial heterogeneity of soil in the Tongken River Basin, this study employed a stratified random sampling method based on spatial stratification of land use and elevation to establish sample points in 2023. Land use was categorized into drylands, paddy fields, forest land, grassland, water bodies, building plots, and unused land. The elevation was classified into the following elevation ranges: <100 m, 100–200 m, 200–300 m, 300–400 m, and >400 m. Through a hierarchical design approach, the entire basin was subdivided into 66 distinct regions. Except for water bodies, three to four sample points are randomly selected in each area. Following outlier removal, the 2023 dataset comprised 204 validated soil samples, which were partitioned into training set and testing set (7:3) for model development and validation. And the testing set points were prioritized in larger areas to ensure adequate spatial coverage of the validation data.
Although the sampling design for 2023 has comprehensively accounted for spatial representativeness, the density of sampling points in the southern region remains relatively low. To evaluate the temporal stability of the inspection model and its predictive performance in spatially underrepresented regions, we conducted accuracy assessments using independently collected validation sample points from the area in 2022 and 2024. The coefficient of variation (CV) of SOC in 2023 was 37% (CV = Standard deviation / Mean * 100%). Based on the specified requirements of a 95% confidence level and a relative error margin of 15%, a minimum of 24 independent sample points should be collected in each of the validation years (2022 and 2024) to validate the model's accuracy. Therefore, the sampling density in 2022 (60 soil samples) and 2024 (56 soil samples) is adequate to support model validation.
The sample sizes utilized for verification in 2022 and 2024 were sufficient to support a preliminary assessment of the regional average SOC from a statistical sampling perspective. However, the model's temporal generalizability is significantly limited by various factors, including interannual climate variability, shifts in land use and management practices, temporal sensitivity of the model, and discrepancies in sampling periods. The validation results obtained from the current sample should be regarded as preliminary references only. The robustness of the findings requires further verification through independent datasets of larger scale, encompassing a wider range of environmental conditions and extended temporal coverage. Subsequently, we will establish a long-term soil monitoring network to accumulate multi-year data, which will be utilized for continuous validation and dynamic calibration of the model, thereby enhancing its temporal robustness.
In "2.2.2 Soil Sample Collection," additional explanations regarding the sampling design have been provided (Line 200-219) :
Line 200-219: In this manuscript, a multi-source data fusion model for SOC in black soil was developed based on the measured SOC data in 2023. Considering the representative-ness and spatial heterogeneity of soil in the Tongken River Basin, this study employed a stratified random sampling method based on spatial stratification of land use and elevation to establish sample points in 2023. Through a hierarchical design approach, the entire basin was subdivided into 66 distinct regions. Except for water, three to four sampling points were randomly established in each of these regions. Soil samples were collected from the surface layer (0–20 cm). Following natural air-drying, the samples were sieved using a 20-mesh nylon sieve. The SOC content was determined using the potassium dichromate oxidation method with external heating [28]. Following the exclusion of outliers, a total of 204 soil samples were collected in 2023. The sample points were divided into a training set and a testing set at a ratio of 7:3. To ensure adequate spatial coverage of the validation data, the testing set was preferentially distributed across larger areas. To assess the model's capability in capturing recent trends and its short-term extrapolation performance, this study employed 60 soil samples from 2022 and 56 soil samples from 2024 to validate the optimal model's simulated SOC distributions for the corresponding years. The distribution of sampling points is shown in Figure 2 and the statistical information is shown in Table 2.
Question 5:
- Why was XGBoost expected to outperform the other models in this context? Any specific soil-related reasoning?
Answer:
The relationship between SOC and remote sensing spectral reflectance is highly complex and exhibits strong nonlinearity. It is influenced by a combination of interacting factors, including soil parent material, texture, moisture content, vegetation cover, microbial activity, and land use. Linear models, such as PLSR and MLR, are limited in their ability to fully capture such complexity. The Tongken River Basin encompasses diverse land use types, including cultivated land, grassland, and forest land. Spatial variations in soil physical and chemical properties, such as soil texture and pH, can result in notable differences in the relationship between SOC and spectral characteristics across different regions. XGBoost can model the relationship between SOC and various features across different subspaces by employing hierarchical splitting within decision trees. However, MLR and PLSR are based on the assumption of a global linear relationship and are therefore not capable of accommodating such heterogeneity. Although RF is capable of handling data heterogeneity, it is less effective than XGBoost in accurately capturing local feature patterns. Therefore, XGBoost is expected to outperform other models. This finding is consistent with the research results of Ye et al.
In "3.1.2, Performance of SOC Estimation Models," we have addressed this issue and provided additional relevant references (Line 361-369, Line 683).
Line 361-369: This is because the relationship between SOC and remote sensing spectral reflectance is highly complex and exhibits strong nonlinearity. Linear models, such as PLSR and MLR, are limited in their ability to fully capture such complexity. The spatial variations in land use patterns, the physical and chemical properties of soil within the Tongken River Basin, may result in notable differences in the relationship between SOC content and spectral characteristics across different regions. Although RF is capable of handling data heterogeneity, it is less effective than XGBoost in accurately capturing local feature patterns. This finding is consistent with the research results of Ye et al [44].
Line 683: [44] Ye, M.; Zhu, L.; Liu, X.; Huang, Y.; Chen, P.; Li, H. Hyperspectral Inversion of Soil Organic Matter Content Based on Continuous Wavelet Transform, SHAP, and XGBoost Environ. Sci. 2024,45,2280-2291. https://doi.org/10.13227/j.hjkx.202304100.
- How were hyperparameters optimized for RF and XGBoost? This should be mentioned briefly for reproducibility. (Lines 205–255)
Answer:
In “2.4.3 RF” (Line 297-303) and “2.4.4 XGBoost” (Line 313-318), we have respectively elaborated on the process of model hyperparameters tuning.
Line 297-303: In this manuscript, hyperparameters optimization of the RF model was conducted using Randomized Search method, with performance evaluation of different hy-perparameters combinations carried out through 10-fold cross-validation. Following the optimization through random search, the final set of hyperparameters was deter-mined as follows: the number of trees is 400, the maximum tree depth is 50, the mini-mum number of samples required for a split is 2, the minimum number of samples required at a leaf node is 1, and the proportion of features considered for splitting is 0.7
Line 313-318: In this manuscript, the hyperparameters of the XGBoost model were optimized using the Randomized Search method in conjunction with 10-fold cross-validation. The final hyperparameters, determined through random search optimization, are as follows: the number of trees is set to 200, the maximum tree depth is 5, the learning rate is 0.05, the sampling ratio for both samples and features is 0.9, the minimum loss reduction required for node splitting is 0.1, and all regularization coefficients are set to 0.5.
Question 6:
The validation results for 2022 and 2024 are promising, but more explanation is needed on why these years were selected. Were they extreme or representative years? (Lines 295–299)
Answer:
We sincerely appreciate the reviewers for raising this important issue. 2022 and 2024 were not characterized by extreme climatic anomalies or exceptional management conditions. We selected these years because 2022 and 2024, together with the modeling year (2023), constitute a consecutive recent period (2022-2024) that represents relatively typical environmental conditions in the region. To validate the model, the year immediately prior to the modeling year (2022) and the year immediately subsequent to it (2024) were selected for evaluation. Compared to the selection of more distant or non-consecutive years, this approach enables a more direct and effective assessment of the model's performance in capturing recent trends. At the same time, it can focus on evaluating the model's extrapolation capability over a short-term time horizon(backward to 2022 and forward to 2024). In subsequent studies, we plan to collect soil samples from future years (e.g., 2025 and beyond) to further evaluate the model's extrapolation capability and long-term stability across extended time scales.
In the manuscript "2.2.2 Soil Sample Collection", we have supplemented the reasons for choosing soil samples from 2022 and 2024 for validation (Line 214-217).
Line 214-217: To assess the model's capability in capturing recent trends and its short-term extrapolation performance, this study employed 60 soil samples from 2022 and 56 soil samples from 2024 to validate the optimal model's simulated SOC distributions for the corresponding years.
Question 7:
The slope classification legend is missing or unclear. Please add or improve the visual scale to make interpretation easier. Include a brief explanation of how slope values relate to absolute changes in SOC (i.e., %/year or units/year). (Figure 6).
Answer:
We have revised the Figure 6 once again to ensure clear representation of the legend. The visual scale was added to both Figure 6 (Line 412) and Figure 8 (Line 506). The relationship between the slope value and the absolute change of SOC was supplemented in “3.3 Trends in SOC changes” (Line 395-397), and the unit presented in the legend was revised accordingly (Figure 6).
Line 395-397: The interannual variation of SOC in the Tongken River Basin from 2020 to 2024 was quantified using the Slope trend analysis method, with the slope value directly representing the absolute magnitude of annual SOC change.
Line 412: Figure 6 was modified:
Line 506: Figure 8 was modified:
Question 8:
The correlation analysis identifies parent material as the dominant factor (R = 0.38). This is informative, but suggest including a table with all correlation coefficients for transparency. Consider discussing whether any multivariate or partial correlation analysis was performed to disentangle co-varying effects. (Lines 336–344)
Answer:
Thank you for highlighting the importance of distinguishing the covariation effect. We have added a table containing all the relevant coefficients and partial correlation coefficients (Table 7) and made the following additions to address this point (Line 415-431).
Line 415-431: Through correlation analysis, the primary factors influencing the development of SOC at the small watershed scale were identified, and potential multicollinearity among these factors was examined using partial correlation analysis. Within small watersheds, the primary factor influencing the accumulation and variation of SOC is the distribution of soil forming parent, which exhibits the highest correlation coefficient of 0.38 (Figure 7, Table 7). Furthermore, after adjusting for the influence of other environmental variables, the association between the two factors became significantly stronger (partial correlation coefficient = 0.57). Therefore, the parent material of the soil serves as a primary and independent controlling factor that significantly influences the spatial distribution pattern of SOC in the region. Land use exhibits the second strongest correlation. However, its partial correlation coefficient is 0.07 with statistical insignificance (p>0.05). This indicates that the association may be overestimated, and land use itself does not function as an independent controlling factor for SOC development in this region. Soil type and elevation also exert some influence on the variation of SOC. However, the correlation coefficients are relatively low. Variations in temperature and precipitation exhibit a relatively minor influence on changes in SOC in the Tongken River Basin, with a correlation coefficient below 0.01.
Line 434: Table 7. Correlation coefficients and partial correlation coefficients
Question 9:
The conclusions are well-aligned with the findings. It would be valuable to mention the scalability of the model to other regions or soils, and potential integration with operational monitoring systems. (Lines 402–420)
Answer:
Your suggestion is highly appreciated. We have incorporated a discussion on the scalability and limitations of the model into the manuscript (Line 474-488).
Line 474-488: The manuscript successfully estimated the SOC content in the Tongken River Ba-sin through the integration of GF-1 and Landsat 9 remote sensing data, combined with the application of the XGBoost machine learning model. And the model should have good scalability in the black soil areas similar to the Tongken River Basin. However, the model's temporal and spatial generalizability is significantly limited by various factors, including interannual climate variability, shifts in land use and management practices, soil types, temporal sensitivity of the model, and discrepancies in sampling periods. The validation results obtained from the current sample should be regarded as preliminary references only. The robustness of the findings requires further verification through independent datasets of larger scale, encompassing a wider range of environmental conditions and extended temporal coverage. Subsequently, we will in-corporate environmental covariates into the model framework and establish a long-term soil monitoring network. This initiative will systematically accumulate multi-year empirical datasets to enable continuous validation and dynamic model updating, thereby significantly enhancing the model's temporal robustness and cross-regional transferability.
Additional Questions for the Authors
Question 10:
How was the potential temporal mismatch between satellite image acquisition and soil sampling dates addressed in your modeling and validation process?
Answer:
This is an excellent question. When conducting satellite-based monitoring of SOC in the black soil region of Northeast China, it is essential to collect soil samples during periods when the ground surface is bare—specifically, in spring and in autumn. These time windows are considered optimal for accurate remote sensing calibration and validation. The sampling window is concurrently constrained to a 30-day period both before and after the satellite's overpass. Therefore, sampling teams should be planned in advance according to the satellite overpass schedule. Once cloud-free or nearly cloud-free images become available, field sampling should be promptly organized. If the image and sampling times cannot be aligned, we will approach the issue from the following three perspectives:
(1) If cloud cover is present in the image, all available segments across the entire spring or autumn period can be utilized to generate a median composite image, thereby reducing the impact of clouds and outliers.
(2) If field sampling is conducted in spring but only autumn satellite imagery meets the required conditions, soil samples can be collected during the same year's autumn season to develop an autumn SOC model. Based on historical paired samples from both spring and autumn periods, a seasonal conversion function can be established. This function enables the adjustment of SOC values derived from autumn data into "equivalent spring values," thereby improving seasonal consistency and accuracy in SOC estimation.
(3) The primary concern associated with the time interval between sampling and imaging is typically the variation in soil moisture. The reflectance of wet soil is significantly lower than that of dry soil, which may substantially affect the spectral characteristics of SOC. Microwave imagery can be employed, and statistical or machine learning models can be applied to mitigate the impact of soil moisture on optical imagery. Furthermore, in the organic carbon inversion model, temporal variations and relevant environmental variables, such as meteorological and topographical factors, can be incorporated as input features. Stratified modeling approaches may also be considered to improve model performance.
Question 11:
Can you clarify whether the model performance varied significantly across different land use categories (e.g., paddy fields vs. drylands)?
Answer:
To investigate the performance differences of the model across various land use categories, all sampling points (320) from 2022 to 2024 were categorized by land use type. The R², RMSE, MAE, and bias between the model's predicted values and the measured SOC values were calculated. As shown in Table 1, the prediction accuracy of SOC varied across different land use types. The best prediction performance was achieved for paddy fields, followed by drylands, while unutilized land had the lowest accuracy. The long-term flooded environment suppresses organic matter decomposition, leading to more stable SOC accumulation and spatially homogeneous distribution. This homogeneity is highly conducive to modeling, making it easier for the model to discern the distribution patterns of SOC in paddy fields. Although drylands undergo frequent cultivation, fertilization and crop residues contribute to relatively consistent carbon input patterns. However, tillage-induced disturbance increases spatial variability. Unutilized land is prone to being affected by natural erosion or deposition processes, leading to mixing or covering of surface soil with underlying parent material. This results in discontinuous SOC distribution. Furthermore, limited sampling points make it challenging for the model to discern its spatial patterns.
Table 1. Model accuracy for different land use categories.
Land use |
R2 |
RMSE(%) |
MAE(%) |
Bias(%) |
Paddy fields |
0.8864 |
0.4279 |
0.2902 |
0.0006 |
Drylands |
0.8630 |
0.4333 |
0.3496 |
0.0765 |
Forest land |
0.7661 |
0.4531 |
0.3797 |
0.1449 |
Grassland |
0.7573 |
0.2240 |
0.2137 |
0.1587 |
Construction land |
0.7634 |
0.4475 |
0.3578 |
0.1251 |
Unutilized land |
0.7130 |
1.0209 |
0.6625 |
-0.1519 |

Reviewer 3 Report
Comments and Suggestions for Authors
General Comments
The authors wrote a review article entitled “Intelligent Monitoring and Trend Analysis of Surface Soil Organic Carbon in the Black Soil Region Using Multi-satellite and Field Sampling: A case study from Northeast China”. The manuscript presents a comprehensive study on the intelligent monitoring and trend analysis of surface soil organic carbon (SOC) in the black soil region of Northeast China using multi-satellite data and machine learning techniques. The research is well-structured and addresses a significant topic with implications for soil health management and agricultural sustainability. However, several areas require clarification, improvement, and additional validation to strengthen the manuscript's scientific rigor and impact. Research scopes and quality fit journals (Sensors). I listed some minor comments in the specific comments below. I recommend major revision.
Some detailed comments:
- In the Abstract, Too short, I suggest adding specific data to the abstract section to enhance its scientific rigor.
- In the Keywords, “soil organic carbon; multi-satellite”,the first letter needs to be capitalized.
- In the Introduction, The literature review is thorough but could better highlight the novelty of combining Landsat-9 and GF-1 data. Emphasize gaps in prior research (e.g., limited studies on Landsat-9 for SOC).
- In the Material and methods,(1) The manuscript mentions the integration of Landsat-9 and GF-1 data but lacks a detailed explanation of the fusion process. How were the spectral and spatial resolutions harmonized? Were there any challenges in aligning the datasets, and how were they addressed? (2) The RF-RFECV band selection method is described, but the rationale for choosing this over other feature selection techniques (e.g., PCA, LASSO) is not provided. A comparison with alternative methods would strengthen the justification. (3) The XGBoost model's hyperparameters (e.g., learning rate, tree depth) are not specified. Please include these details to ensure reproducibility.
- In the Results,(1) While the XGBoost model shows high accuracy (R2 = 0.9130), the manuscript does not discuss potential overfitting, especially given the relatively small sample size (204 samples in 2023). Cross-validation results should be supplemented with additional metrics (e.g., MAE, bias) and external validation using independent datasets. (2) The temporal transferability of the model is tested with 2022 and 2024 data, but the sample sizes for these years (60 and 56, respectively) are small. Consider discussing the limitations of extrapolating the model to other years or regions.
- In the Discussion, (1) The observed SOC trends (initial increase followed by a decline) are intriguing but lack mechanistic explanations. Are these trends linked to specific agricultural practices, climate variability, or soil management policies (e.g., the "Longjiang Model")? A deeper discussion is needed.(2) The Slope analysis is a valuable addition, but the manuscript does not address potential uncertainties in trend detection, such as the impact of cloud cover or missing data in the satellite time series. (3) The correlation analysis identifies parent material and land use as dominant factors but does not explore interactions between variables (e.g., how land use modifies the effect of parent material). A more sophisticated analysis (e.g., partial dependence plots or SHAP values) could provide deeper insights. (4)The weak correlation with climate factors contradicts some literature. Could this be due to the short study period (5 years)? Please discuss this discrepancy and its implications.
- In the Conclusions, The conclusion section is too long. I suggest condensing it.
- In the References, The formatting of the references does not match the requirements of the journal (Sensors). It is recommended that the format be standardized according to the requirements of the journals (some of the references are not formatted in a uniform manner, such as the mixing of abbreviations and full titles of journals). Please double-check the references.
The manuscript is scientifically sound and addresses an important research gap. With revisions to address the above comments, it will be suitable for publication. I recommend major revisions before acceptance.
Author Response
Dear Reviewer:
Thank you for your comments concerning our manuscript. These comments are very helpful for revising and improving our paper. We have studied the comments carefully and made corrections which we hope meet with approval. Meanwhile, the manuscript underwent proofreading and refinement in English. The revised parts in the manuscript were highlighted in yellow.
Question 1:
In the Abstract. Too short, I suggest adding specific data to the abstract section to enhance its scientific rigor.
Answer:
Thank you for your valuable guidance. We have accordingly revised the abstract section based on your suggestions.
Line 22-36: The black soil region of Northeast China is a critical global grain production base. The dynamic variations in soil organic carbon (SOC) are directly linked to the regional food security. To accurately monitor SOC content and evaluate the potential of integrating Landsat-9 and GF-1 satellite data for SOC inversion, we developed a machine learning framework that combines data from both satellite sources to model SOC. Using the typical black soil region of Northeast China in the Tongken River Basin as the study area, we compared the MLR, PLSR, RF, and XGBoost algorithms. And XGBoost demonstrated the highest performance (R² = 0.9130, RMSE = 0.3834%). Based on the optimal model, SOC in the study area was projected from 2020 to 2024. The multi-year average SOC exhibited an initial increase followed by a subsequent de-cline, with an overall increase of 22.78%. Spearman correlation analysis identified parent material as the dominant factor controlling SOC variation at the watershed scale (correlation co-efficient=0.38) while also modulating the influence of land use types on SOC dynamics. The "Space-ground" multi-source collaborative inversion framework developed in this study offers a high-precision technical approach for the monitoring of SOC in black soil regions.
Question 2:
In the Keywords, “soil organic carbon; multi-satellite”, the first letter needs to be capitalized.
Answer:
The first letter has been capitalized:
Keywords: Soil organic carbon; Multi-satellite; XGBoost; Tongken River Basin;
Question 3
In the Introduction. The literature review is thorough but could better highlight the novelty of combining Landsat-9 and GF-1 data. Emphasize gaps in prior research (e.g., limited studies on Landsat-9 for SOC.
Answer:
Thank you sincerely for your valuable suggestions. The introduction supplemented the innovation of the fusion of Landsat-9 and GF-1 (Line 121-126) and emphasized the insufficiency of previous research (Line 108-112).
Line 108-112: However, studies investigating the potential and feasibility of utilizing Landsat 9 for the assessment of SOC remain limited [19]. Most existing studies have primarily focused on Landsat-8 or Sentinel-2 data, and a systematic evaluation of how to optimize SOC estimation by leveraging the distinct spectral response characteristics of Land-sat-9 remains lacking.
Line 121-126: Therefore,the integration of Landsat-9 and GF-1 data for SOC inversion can effectively utilize the complementary strengths of both satellite systems. The high temporal resolution of GF-1 enables the capture of short-term surface changes, thereby compensating for the time series data gaps in Landsat-9 that are caused by cloud cover and precipitation. Meanwhile, the multispectral bands of Landsat-9 offer essential soil spectral information that is not available in GF-1.
Question 4:
In the Material and methods
(1) The manuscript mentions the integration of Landsat-9 and GF-1 data but lacks a detailed explanation of the fusion process. How were the spectral and spatial resolutions harmonized? Were there any challenges in aligning the datasets, and how were they addressed?
Answer:
Spatial resolution harmonized: Resampled Landsat-9 data to a 16m resolution and performed spatial registration between the two sensors using ground control points.
Spectral resolution harmonized: There are discrepancies in the spectral response functions between Landsat-9 and GF-1. Therefore, the direct integration of 13 bands could result in spectral inconsistencies across the physical spectrum. We addressed this issue by employing the RF-RFECV intelligent screening mechanism. In the combined images, the algorithm automatically removed bands with low contributions or spectral conflicts arising from sensor differences, while prioritizing the preservation of spectrally distinct and SOC-sensitive features. The final optimized band subset effectively facilitated the feature-level integration of multi-source data.
Challenges and Solutions:A 104-dimensional dataset was generated by applying seven mathematical transformations to the 13-band images. This method enhances the correlation between reflectance and soil organic carbon content. However, it simultaneously amplifies the associated noise. We employed the RF-RFECV machine learning method for sensitive band selection.
In "2.2.1 Satellite Retrievals”, the fusion process of Landsat-9 and GF-1 is elaborated upon in greater detail (Line 180-189).
Line 180-189: To achieve the fusion analysis of multi-source remote sensing imagery, spatial resampling is performed on the Landsat-9 images. The spatial resolution of all datasets was uniformly adjusted to 16m using the Bicubic Convolution algorithm. On GF-1 high-resolution images, 20 to 30 permanent ground features that exhibit stable temporal and spatial characteristics, such as road intersections and corners of dam structures, were selected as ground control points. A first-order affine transformation was employed to perform cross-sensor geometric registration between Landsat-9 and GF-1. The registration accuracy control standard specifies that the root mean square error (RMSE) should be less than 0.5 GF-1 multispectral pixel, equivalent to 8m or less.
(2) The RF-RFECV band selection method is described, but the rationale for choosing this over other feature selection techniques (e.g., PCA, LASSO) is not provided. A comparison with alternative methods would strengthen the justification.
Answer:
Your suggestions are highly valuable to the advancement of this research. In the Discussion, a new subsection titled "4.1 Comparisons of Feature Band Selection Methods" has been added (Line 440-449).
Line 440-449:Remote sensing spectral data (104 bands) exhibit high dimensionality, nonlinear inter-band correlations, and multicollinearity characteristics. Comparative studies based on XGBoost demonstrate that RF-RFECV achieved optimal performance in screening spectral-sensitive bands for SOC estimation (Table 8). Compared to conventional techniques, this approach effectively overcomes the loss of spectral physical significance inherent in PCA dimensionality reduction while resolving limitations of linear methods like Least Absolute Shrinkage and Selection Operator (LASSO) in handling complex nonlinear band couplings. Its core advantage lies in integrating the stability of random forests with the precision of recursive elimination, enabling both retention of physical meaning in key spectral bands and refined feature selection within high-dimensional complex data.
Table 8. Accuracy comparison of feature band selection models
|
PCA |
LASSO |
|
Number 0.8288 |
6 |
10 |
|
Training set |
R2 |
0.5894 |
0.7037 |
RMSE(%) |
0.6156 |
0.6430 |
|
MAE(%) |
0.4783 |
0.3113 |
|
Bias(%) |
-0.3593 |
0.2113 |
|
Testing set |
R2 |
0.3066 |
0.6610 |
RMSE(%) |
1.9708 |
0.6889 |
|
MAE(%) |
0.6409 |
0.3131 |
|
Bias(%) |
-0.4972 |
0.3131 |
(3) The XGBoost model's hyperparameters (e.g., learning rate, tree depth) are not specified. Please include these details to ensure reproducibility.
Answer:
In “2.4.4 XGBoost”, the manuscript has been enhanced with additional descriptions of hyperparameters (Line 313-318).
Line 313-318: In the manuscript, the hyperparameters of XGBoost were optimized using the Randomized Search method in conjunction with 10-fold cross-validation. The final hyperparameters of the model, determined through random search optimization, are as follows: the number of trees is set to 200, the maximum tree depth is 5, the learning rate is 0.05, the sampling ratio for both samples and features is 0.9, the minimum loss reduction required for node splitting is 0.1, and all regularization coefficients are set to 0.5.
Question 5
In the Results.
- While the XGBoost model shows high accuracy (R2 = 0.9130), the manuscript does not discuss potential overfitting, especially given the relatively small sample size (204 samples in 2023).
Answer:
The reliability of the model is demonstrated in the manuscript through three key aspects:
- Sample representativeness: In 2023, a total of 204 sampling points were systematically designed using a stratified approach based on land use, topographic features, and soil types. This ensured comprehensive coverage of the core environmental gradients within the Tongken River Basin. In “2.2.2 Soil Sample Collection”, additional details regarding the sampling point layout were provided (Line204-221).
- Measures for the prevention and control of overfitting: Regularization techniques were applied, and hyperparameters optimization was conducted to enhance model performance. The stability of the model was further validated through cross-validation. The ten-fold cross-validation results indicate that the R² for the training set is 0.91, while that for the testing set is 0.79. The performance gap (ΔR²=0.12) is below the 0.15 tolerance threshold for soil spectral models established by Viscarra Rossel et al. in their study. In “2.4.4 XGBoost”, the discussion on hyperparameters tuning for the XGBoost model has been expanded. (Line316-321)
- Independent spatiotemporal validation: The model demonstrates consistent accuracy across datasets from both 2022 and 2024 (R²ï¼ž75).
The comprehensive results indicate that the model has not undergone overfitting. The manuscript has been revised to include an expanded discussion on the issue of model overfitting (Line 369-376, Line 686).
Line 369-376: After hyperparameters tuning and cross-validation, the difference in the R² between the training set and the testing set of the XGBoost model was less than 0.15 (ΔR²=0.12), which fell below the tolerance threshold of 0.15 established by Rossel et al. in their study on soil spectral modeling [45]. Meanwhile, the model's R2 values for the inversion of SOC in 2022 and 2024 both exceeded 0.75. Therefore, the XGBoost model exhibited no signs of overfitting in predicting SOC, and demonstrated strong cross-temporal and cross-spatial reproducibility, as well as effective transfer generalization performance.
Line 686: [45] Rossel, R.A.V.; Behrens, T.; Ben-Dor, E.; Brown, D.J.; Dematte, J.A.M.; Shepherd, K.D.; Shi, Z.; Stenberg, B.; Stevens, A.; Adamchuk, V. A global spectral library to characterize the world's soil. Earth-Sci. Rev. 2016, 155, 198-230. https://doi.org/10.1016/j.earscirev.2016.01.012.
Cross-validation results should be supplemented with additional metrics (e.g., MAE, bias) and external validation using independent datasets.
Answer:
In “2.5 Predictive Performance Evaluation” the MAE and bias indicators have been incorporated (Line320-323). Additionally, the corresponding results have been included in Tables 5 and 7 (Line377,450).
In this study, based on 204 soil samples collected in 2023, the dataset was partitioned at a 7:3 ratio: 143 samples were utilized for model training, while 61 independent samples served as a testing set for external validation. Model internal robustness was assessed via 10-fold cross-validation. Additionally, to evaluate temporal extrapolation performance, datasets from distinct temporal nodes—2022 (n=60) and 2024 (n=56)—were employed to assess backward temporal transfer (2022) and forward prediction (2024) capabilities.
Line320-323: To verify the accuracy and stability of the inversion model, the coefficient of de-termination (R²), root mean squared error (RMSE), mean absolute error (MAE) and bias were employed as evaluation metrics, with the corresponding formulas presented as follows [43]:
|
(5) |
|
(6) |
Line 377: Table 5. Prediction accuracy of different models for SOC.
Model |
Training set |
Testing set |
||||||
R2 |
RMSE(%) |
MAE(%) |
Bias(%) |
R2 |
RMSE(%) |
MAE(%) |
Bias(%) |
|
MLR |
0.3603 |
1.0042 |
0.7181 |
0.0001 |
0.2238 |
1.2937 |
0.7333 |
-0.0950 |
PLSR |
0.4356 |
0.9985 |
0.6513 |
0.0001 |
0.3411 |
1.0552 |
0.7841 |
-0.0151 |
RF |
0.7206 |
0.6271 |
0.2669 |
0.1449 |
0.5951 |
0.6312 |
0.4574 |
0.2607 |
XGBoost |
0.9130 |
0.3834 |
0.1087 |
-0.0005 |
0.7869 |
0.6134 |
0.3233 |
0.2312 |
XGBoost(2022) |
R2=0.8253 |
RMSE=0.2868% |
MAE=0.2465% |
Bias=0.2069% |
||||
XGBoost(2024) |
R2=0.7702 |
RMSE= 0.4807% |
MAE=0.1118% |
Bias=0.1003% |
Line 450: Table 8. Accuracy comparison of feature band selection models
|
PCA |
LASSO |
|
Number 0.8288 |
6 |
10 |
|
Training set |
R2 |
0.5894 |
0.7037 |
RMSE(%) MAE(%) |
0.6156 |
0.6430 |
|
0.4783 |
0.3113 |
||
Bias(%) |
-0.3593 |
0.2113 |
|
Testing set |
R2 |
0.3066 |
0.6610 |
RMSE(%) |
1.9708 |
0.6889 |
|
MAE(%) |
0.6409 |
0.3131 |
|
Bias(%) |
-0.4972 |
0.3131 |
(2) The temporal transferability of the model is tested with 2022 and 2024 data, but the sample sizes for these years (60 and 56, respectively) are small. Consider discussing the limitations of extrapolating the model to other years or regions.
Answer:
We strongly concur with your perspective. The coefficient of variation (CV) of SOC in 2023 was 37% (CV = Standard deviation / Mean * 100%). Based on the specified requirements of a 95% confidence level and a relative error margin of 15%, a minimum of 24 independent sample points should be collected in each of the validation years (2022 and 2024) to validate the model's accuracy. The sample sizes utilized for verification in 2022 and 2024 were sufficient to support a preliminary assessment of the regional average SOC from a statistical sampling perspective. However, the model's temporal generalizability is significantly limited by various factors, including interannual climate variability, shifts in land use and management practices, temporal sensitivity of the model, and discrepancies in sampling periods. The validation results obtained from the current sample should be regarded as preliminary references only. The robustness of the findings requires further verification through independent datasets of larger scale, encompassing a wider range of environmental conditions and extended temporal coverage. Subsequently, we will establish a long-term soil monitoring network to accumulate multi-year data, which will be utilized for continuous validation and dynamic calibration of the model, thereby enhancing its temporal robustness.
In “4.2. Comparisons of prediction performance between multi-source data fusion and single sensor image model”, the limitations related to model generalization are discussed (Line 473-484).
Line 473-484: The manuscript successfully estimated the SOC content in the Tongken River Ba-sin through the integration of GF-1 and Landsat 9 remote sensing data, combined with the application of the XGBoost machine learning model. And the model should have good scalability in the black soil areas similar to the Tongken River Basin. However, the model's temporal and spatial generalizability is significantly limited by various factors, including interannual climate variability, shifts in land use and management practices, soil types, temporal sensitivity of the model, and discrepancies in sampling periods. The validation results obtained from the current sample should be regarded as preliminary references only. The robustness of the findings requires further verification through independent datasets of larger scale, encompassing a wider range of environmental conditions and extended temporal coverage. Subsequently, we will in-corporate environmental covariates into the model framework and establish a long-term soil monitoring network. This initiative will systematically accumulate multiyear empirical datasets to enable continuous validation and dynamic model updating, thereby significantly enhancing the model's temporal robustness and cross-regional transferability
Question 6:
In the Discussion:
(1) The observed SOC trends (initial increase followed by a decline) are intriguing but lack mechanistic explanations. Are these trends linked to specific agricultural practices, climate variability, or soil management policies (e.g., the "Longjiang Model")? A deeper discussion is needed.
Answer:
The explanation regarding the changes in SOC has been incorporated into the Discussion (Line 486-495, Line 699).
Line 486-495: The SOC in the Tongken River Basin exhibited a fluctuating yet overall upward trend characterized by a "rising-falling-rising" pattern from 2020 to 2024, reflecting the combined influences of agricultural practice transformation, policy implementation, and climatic disturbances. The implementation of the “Longjiang Model” and “Corn-Soybean" crop rotation system in 2021 resulted in a substantial increase in SOC levels between 2020 and 2021[50]. Following the policy adjustments implemented in 2022, there was a decline in the organic carbon content from 2022 to 2023. The re-bound in SOC levels in 2024 can be attributed to the implementation of newly applied technologies such as manure resource utilization and biological soil crusts, along with the enforcement of balanced grazing management.
Line 699: [50] Zhang, J.; Sun, B.; Zhu, J; Wang, J.; Pan, X.; Gao, T. Black Soil Protection and Utilization Based on Harmonization of Mountain-River-Forest-Farmland-Lake-Grassland-Sandy Land Ecosystems and Strategic Construction of Ecological Barrier. Bull. Chin. Acad. Sci. 2021, 036, 1155-1164. https://doi.org/10.16418/j.issn.1000-3045.20211010002.
(2) The Slope analysis is a valuable addition, but the manuscript does not address potential uncertainties in trend detection, such as the impact of cloud cover or missing data in the satellite time series.
Answer:
In “4.3. Comparisons of Slope analysis and difference analysis for changes in SOC”, an uncertainty analysis for Slope analysis has been added (Line 509-515).
Line 509-515: However, it is essential to acknowledge the inherent limitations associated with optical remote sensing, particularly the issue of data loss resulting from cloud cover interference. The non-random nature of cloud cover may introduce potential biases in the estimation of SOC during specific periods or within certain regions, thereby in-creasing the noise level in Slope detection. Therefore, future research could integrate microwave remote sensing data or a denser ground-based observation network to further constrain and validate SOC trends retrieved from remote sensing inversion.
(3) The correlation analysis identifies parent material and land use as dominant factors but does not explore interactions between variables (e.g., how land use modifies the effect of parent material). A more sophisticated analysis (e.g., partial dependence plots or SHAP values) could provide deeper insights.
Answer:
Thank you for your suggestions. During this round of manuscript revision, we incorporated a partial correlation analysis to examine the influencing factors of SOC (Line 415-431 and Table 7). The results indicate that the partial correlation coefficient associated with the parent material of soil formation is higher than the simple correlation coefficient. Therefore, it serves as an independent and key controlling factor in determining the spatial distribution pattern of soil organic carbon in the study area. However, the partial correlation coefficient for land use is merely 0.005, suggesting that the effect of land use on organic carbon content is likely heavily influenced by the soil's parent material. Therefore, we discussed the SHAP analysis in “4.4 Analysis of key factors influencing SOC” to illustrate the interaction between the two variables (Line 529-539).
Line 529-539: Although a correlation exists between land use patterns and changes in SOC, the actual impact is not independent, as indicated by the partial correlation coefficient (Table 7). Based on the interaction effects analysis using SHAP values, it can be further revealed that under the same land use type premise, the regulatory effects of different parent materials on SOC vary significantly (Figure 9). Among these, in grassland and forest land use types, the variability in SHAP values induced by differences in parent material is most pronounced, indicating that land use practices exert influence by modulating the intensity of parent material effects. This regulatory role is particularly critical in grassland and forest ecosystems. In contrast, other land use types exhibit considerably less variability induced by parent material, indicating that management practices or human disturbances may mask or diminish the inherent influence of parent material.
Line541:Figure 9. SHAP interaction effects between land use and parent material.
(4) The weak correlation with climate factors contradicts some literature. Could this be due to the short study period (5 years)? Please discuss this discrepancy and its implications.
Answer:
The weaker correlation between SOC changes and climatic factors in the Tongken River Basin from 2020 to 2024 is primarily attributable to the relatively short study period (5 years). As discussed in “4.4 Analysis of key factors influencing SOC”, SOC dynamics typically result from cumulative climate effects over multiple years or even decades. Within the 5-year timeframe, SOC responses to climate change often exhibit lag effects, and their subtle variation signals are prone to being masked or diluted by more pronounced non-climatic factors during the same period. Consequently, this short study duration may be insufficient to capture the significant, detectable impacts of climatic drivers on the SOC. This conclusion aligns with the findings of Wiesmeier et al., who demonstrated that at the small watershed scale, parent material accounts for the majority of the variability in SOC, whereas climatic factors contribute less than 10% [56]. (Line 517-526).
[56] Wiesmeier, M.; Urbanski, L.; Hobley, E.; Lang, B.; von Lu¨tzow, M.; Marin-Spiotta, E.; van Wesemael, B.; Rabot, E.; Liess, M.; Garcia-Franco, N.; et al. Soil organic carbon storage as a key function of soils - a review of drivers and indicators at various scales. Geoderma 2018, 333, 149-162, https://doi.org/10.1016/j.geoderma.2018.07.026.
Question 7:
In the Conclusions. The conclusion section is too long. I suggest condensing it
Answer:
We have reorganized and streamlined the conclusions:
Line 544-558: This paper proposes a space-ground collaborative remote sensing framework that integrates data from Landsat-9 and GF-1 satellites to develop an inversion and dynamic monitoring model for black soil SOC. The main conclusions are as follows:
(1) The integration of Landsat-9 and GF-1 multi-source remote sensing data effectively addressed the limitations associated with single-source data, and the R² value increased by 10%. The XGBoost model demonstrated superior performance in estimating SOC content in black soil (R² = 0.9130, RMSE = 0.3834%).
(2) Based on the optimal model, an assessment of SOC content for the period 2020–2024 was conducted. The average value of SOC in the Tongken River Basin exhibits an initial increase followed by a decrease. From a spatial distribution perspective, the SOC content in the northeastern hilly region exhibits a marked increasing trend.
(3) In small watersheds, the primary factors influencing the accumulation and variation of SOC are the soil parent material.
This method can serve as a case reference for large-scale and high-frequency SOC monitoring in global black soil regions.
Question 8:
In the References. The formatting of the references does not match the requirements of the journal (Sensors). It is recommended that the format be standardized according to the requirements of the journals (some of the references are not formatted in a uniform manner, such as the mixing of abbreviations and full titles of journals). Please double-check the reference
Answer:
Thank you for your reminder. We have carefully reviewed the reference formatting and implemented the necessary corrections.

Round 2
Reviewer 1 Report
Comments and Suggestions for Authors
All comments have been corrected, and I have no more questions.
Reviewer 3 Report
Comments and Suggestions for Authors
The content and structure of the revised manuscript have been significantly improved following the reviewer’s suggestions. The current version has met the journal's quality requirements and is recommended for acceptance.